# Attention Satisfies: A Constraint-Satisfaction Lens on Factual Errors of Language Models

**Mert Yuksekgonul**[†‡]
Stanford University

**Varun Chandrasekaran**[‡]
University of Illinois Urbana-Champaign

**Erik Jones**[‡]
UC Berkeley

**Suriya Gunasekar**   **Ranjita Naik**   **Hamid Palangi**   **Ece Kamar**   **Besmira Nushi**[†]
Microsoft Research

## Abstract

We investigate the internal behavior of Transformer-based Large Language Models (LLMs) when they generate factually incorrect text. We propose modeling factual queries as constraint satisfaction problems and use this framework to investigate how the LLM interacts internally with factual constraints. We find a strong positive relationship between the LLM's attention to constraint tokens and the factual accuracy of generations. We curate a suite of 10 datasets containing over 40,000 prompts to study the task of predicting factual errors with the Llama-2 family across all scales (7B, 13B, 70B). We propose SAT Probe, a method probing attention patterns, that can predict factual errors and fine-grained constraint satisfaction, and allow early error identification. The approach and findings take another step towards using the mechanistic understanding of LLMs to enhance their reliability.[1]

## 1    Introduction

Large language models (LLMs) encode substantial knowledge (Petroni et al., 2019; Srivastava et al., 2022), yet they are prone to generating factually incorrect text. For instance, LLMs can generate confident-appearing completions with *hallucinations* (Zhang et al., 2023; Ji et al., 2023), fabricating entities or factual claims. As LLMs reach wider audiences and are used for safety-critical applications, understanding the factuality of generations rises to paramount importance. However, our understanding of how LLMs process factual queries and produce errors is nascent. Existing approaches fall into two categories: i) treat the LLM as a black box and query it about generated factual claims, or ii) use white-box methods to study how LLMs internally process factual queries.

Black-box approaches center on analyzing the consistency of the claims of an LLM using follow-up questions with other LLMs (Cohen et al., 2023) or having the LLM self-critique (Zhang et al., 2023; Manakul et al., 2023). However, explanations from LLMs are suggested to be unreliable (Turpin et al., 2023) or to convey contradictory signals, e.g., LLMs can produce an answer and then acknowledge that it is wrong (Zhang et al., 2023; Mündler et al., 2023). Further, black-box methods typically involve multiple generations from LLMs, which may be prohibitively expensive to use in practice.

Mechanistic white-box approaches investigate the internal mechanisms of LLMs to dissect factual recall. For instance, Meng et al. (2022); Geva et al. (2023) focus on facts with the (subject, relation, object) structure (e.g., Paris, capital of, France) and propose insightful mechanisms of how an LLM recalls a fact. They suggest that the multi-layer perceptron layers store facts, and attention layers transfer factual information from the subject tokens. However, these works focus on when the LLM generates factually correct responses. Mechanisms leading to factual errors are scarcely explored.

**Our Contributions:** We investigate the internal mechanisms of LLMs, specifically the attention patterns, when they produce factual errors. We propose to view factual queries as constraint satisfaction

---

[1]Our datasets, evaluation protocol, and methods will be released at https://github.com/microsoft/mechanistic-error-probe.

[†]Correspondence to merty@stanford.edu and besmira.nushi@microsoft.com.

[‡]This work was done while at Microsoft Research.

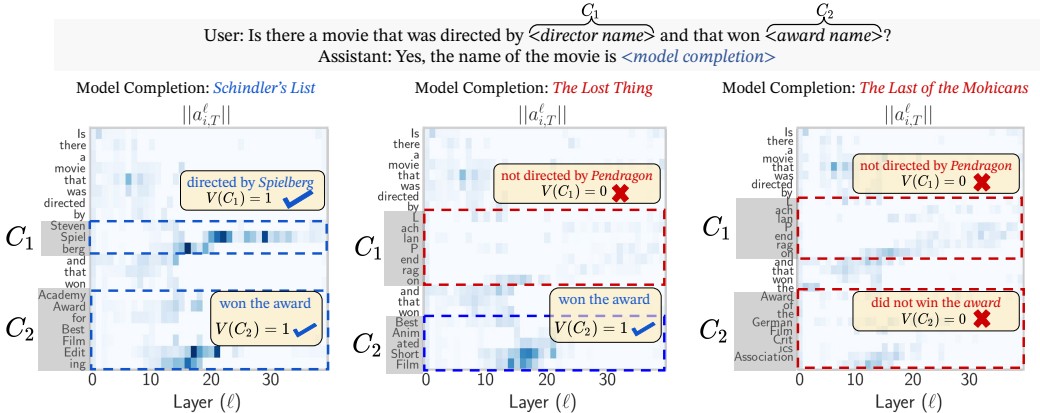

Figure 1: **Tracking attention to predict constraint satisfaction and factual errors.** We consider factual queries to LLMs as constraint satisfaction problems i.e., factual queries impose a set of constraints that the LLM's responses must satisfy. To predict constraint satisfaction (i.e., factual correctness), we track the attention to the constraint tokens in an LLM (here, Llama-2 13B). We find that attention to the constraint tokens highly correlates with factual correctness. The red (resp. blue) text indicates factually incorrect (resp. correct) completions.

problems (CSPs), where queries comprise constraints that completions should satisfy to be factually correct (§3); e.g., in Figure 1 the *director name* or the *award name* are constraints on the model's response to a search query for a movie. We investigate the interaction between constraints, attention, and factual correctness (§4). Our key finding is that attention to constraint tokens correlates with LLM's factual correctness, where less attention to constraints is associated with inaccurate responses. Building on our insights, we propose SAT PROBE, a method that predicts constraint satisfaction (and thus factual correctness), using a simple probe of the LLM's attention to constraints (§5). To test SAT PROBE, we curate a suite of 10 datasets of single- and multi-constraint queries with in total over $40,000$ prompts. We find that SAT PROBE performs comparably to the LLM's confidence. Further, SAT PROBE can predict factual errors halfway through the forward pass to stop the computation partway and save costs. Our findings contribute to the mechanistic understanding of LLMs and demonstrate the potential of model internals to understand and mitigate factual errors.

## 2 BACKGROUND: LANGUAGE MODELS AND FACTUAL RECALL

Our presentation of the Transformer architecture (Vaswani et al., 2017) largely follows that of Meng et al. (2022); Geva et al. (2023); Elhage et al. (2021). For brevity, we omit the details around layer normalization. Assume an input sequence of $T$ tokens $t_1, ..., t_T$ and $t_i \in \mathcal{V}$ for a fixed vocabulary $\mathcal{V}$. A token $t_i$ is represented initially with a $d$-dimensional vector $\mathbf{x}_i^0 \in \mathbb{R}^d$ using an embedding matrix $E \in \mathbb{R}^{|\mathcal{V}| \times d}$. We use $\mathcal{V}^+$ to denote a sequence of tokens.

The architecture consists of $L$ layers that transform the input token embeddings to a sequence of hidden states $\mathbf{x}_1^\ell, \ldots, \mathbf{x}_T^\ell$ at each layer $\ell$ where $\mathbf{x}_i^\ell$ denotes the state of token $i$. Often, each hidden state vector has the same number of dimensions, i.e., $\forall i, \ell \ \mathbf{x}_i^\ell \in \mathbb{R}^d$. The states are obtained by

$$\mathbf{x}_i^\ell = \mathbf{x}_i^{\ell-1} + \mathbf{a}_i^\ell + \mathbf{m}_i^\ell, \tag{1}$$

where we call $\mathbf{m}_i^\ell$ the *MLP contribution* and $\mathbf{a}_i^\ell$ the *attention contribution* to a token $i$ at layer $\ell$, respectively. The LLM produces a predicted probability distribution for the next token $\hat{\mathbb{P}}(t_{T+1}|t_{1:T})$ using a linear softmax layer on the last layer representation $x_T^L$. In this work, we study the interactions among tokens. Since the MLP layers (see Appendix A) in standard Transformers do not capture token interactions, we focus primarily on the attention operation.

The **attention** operation updates each token's state using the previous states at all positions, i.e., the representation for a token is updated by 'attending' to all other tokens. Formally, the operation involves four projection matrices $W_Q^\ell, W_K^\ell, W_V^\ell, W_O^\ell \in \mathbb{R}^{d \times d}$ that correspond to the 'query', 'key', 'value', and 'output' projections. Each of these is split into multiple heads, where $W_Q^{\ell,h}, W_K^{\ell,h}, W_V^{\ell,h} \in \mathbb{R}^{d \times d_h}$ and $W_O^{\ell,h} \in \mathbb{R}^{d_h \times d}$ denote the matrices for head $h$, $H$ is the total number of heads, $d_h$ is the

dimensionality for each head for $h \in [H]$. Often, embeddings are split into equal parts such that $d_h = \frac{d}{H}$ (Elhage et al., 2021; Dar et al., 2022; Touvron et al., 2023). The *attention contribution* from the token $j$ to token $i$, $\mathbf{a}_{i,j}^{\ell}$, is defined as:

$$\mathbf{a}_{i,j}^{\ell} = \sum_{h=1}^{H} A_{i,j}^{\ell,h}(x_j^{\ell-1}W_V^{\ell,h})W_O^{\ell,h}, \qquad (2)$$

$$A^{\ell,h} = \text{Softmax}\left(\frac{\left(X^{\ell-1}W_Q^{\ell,h}\right)\left(X^{\ell-1}W_K^{\ell,h}\right)^T}{\sqrt{d_h/H}}\right), \qquad (3)$$

where $\mathbf{a}_i^l = \sum_{j \in [T]} \mathbf{a}_{i,j}^l$ and Softmax is taken row-wise. $A^{\ell,h} \in \mathbb{R}^{T \times T}$ are the *attention weights* computed by the $h$-th attention head at layer $\ell$, and $A_{i,j}^{\ell,h}$ is the entry in the $i$-th row and $j$-th column of the matrix. For autoregressive LLMs, $A^{\ell,h}$ is lower triangular since each token can only attend to the representation of the previous tokens. For brevity, we use $[H]$ to denote the sequence of integers from 1 to $H$, and superscript $[H]$ indicates stacking items for all $h \in [H]$, i.e., $A_{i,j}^{\ell,[H]} = \{A_{i,j}^{\ell,h}\}_{h=1}^{H} \in \mathbb{R}^H$.

**Mechanics of Factual Recall in Language Models:** Recent work investigates the internal activations of LLMs to understand the mechanics of factual recall. By studying factual queries of the form (subject, relation, object), Meng et al. (2022); Geva et al. (2021) provide evidence that MLP layers store factual associations and Geva et al. (2023); Meng et al. (2022); Elhage et al. (2021) suggest that attention layers transfer factual knowledge to where it will be used. As an example, when an LLM is given the prompt *LeBron James professionally plays*, the information *LeBron James professionally plays basketball* is extracted by the MLP contribution to the tokens for *LeBron James* (subject). Next, the attention layers transfer the information from the tokens of the subject to the last token for the LLM to generate *basketball* (object). While these works mostly study internal mechanisms when the LLM's completions are factually correct, we focus on *when LLMs produce factually incorrect text*.

## 3    FACTUAL QUERIES AS CONSTRAINT SATISFACTION PROBLEMS

Choosing the right framework to study factual errors is challenging. One can naively categorize completions as factually correct or incorrect, yet this binary view can fall short. For example, queries that are easy for the LLM and ones that it barely gets right are indistinguishable since both are labeled as 'correct'. Further, binary labeling prevents us from building an understanding of why some queries are more difficult than others or which parts of the queries drive the LLM to failure.

To systematically study factual queries and LLMs' internal behavior, we propose taking a CSP view:

**Definition 3.1** (Factual Query as a CSP). A factual query is specified by a set of constraints $\mathcal{C} = \{(C_1, V_1), \ldots (C_K, V_K)\}$ where $C_k \in \mathcal{V}^+$ indicates the sequence of tokens for the constraining entity $k$[2], and $V_k : \mathcal{V}^+ \to \{0, 1\}$ is a *verifier* that takes a set of generation tokens as the input and returns whether the constraint indexed by $k$ is satisfied. Under this view, we call a completion $Y$ as a *factual error* if $\exists k \in [K] : V_k(Y) = 0$, that is, if there is a constraint in the factual query that the response does not satisfy[3]. Otherwise, we call the response *factually correct*.

A large set of factual queries can be viewed as a set of constraints that responses must satisfy to be correct, e.g., see Figure 1. This structure is comprehensive; for example, an important subset of queries made by users to search engines has historically been conjunctions of constraints (Spink et al., 2001). Structured and multi-constraint queries are also inherent to faceted search and information retrieval (Tunkelang, 2009; Hahn et al., 2010). Further, under this definition, prior (subject, relation, object) queries (Meng et al., 2022) can be seen to have a single-constraint structure. Similarly, instructions to LLMs are also constraints for controlling the output (Ouyang et al., 2022).

Focusing on the constraints of a CSP can help us reason about the difficulty of a query and the behavior of the LLM. While there are several factors that can influence LLM's behavior, we shall start

---

[2]While it may be nontrivial to generally isolate tokens for the constraining entity for arbitrary queries, in our evaluations we investigate settings in which we assume we have access to this set.

[3]Here, we focus on conjunctions of constraints, but the framework can be generalized to other operations.

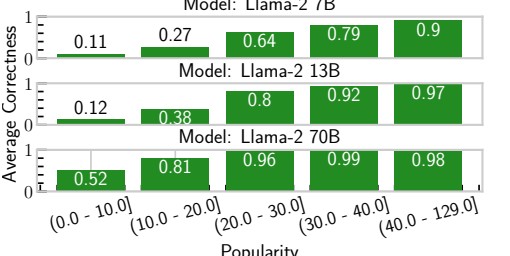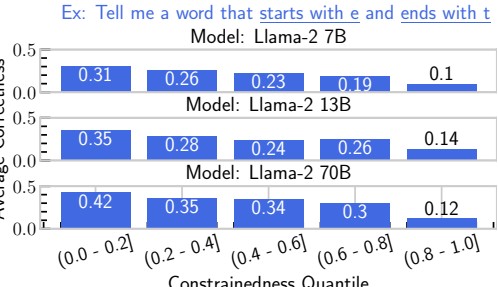

Figure 2: **Difficulty of the factual query vs LLM performance. Left: Popularity vs Correctness** We observe that the more popular the entity in the factual query is, the more correct the LLMs are. **Right: Constrainedness vs Correctness** We observe that the more constrained the problem is (i.e., has a smaller set of potential solutions), the less correct the LLMs are.

with two factors that can describe the difficulty of factual queries: i) the popularity of the constraining entity, and ii) the constrainedness of the query.

**Popularity of the Entity vs LLM Performance:** Recent work has shown the correlation between training data frequency and memorization in LLMs (Carlini et al. (2022); Biderman et al. (2023); *inter alia*). However, even for many open-source LLMs, we cannot compute the frequency of facts since we do not have the training data or a trivial way to search for complex facts. As an accessible proxy for entities from WikiData, we use the number of site links on a page as the *popularity* of entity; we hypothesize that it strongly correlates with the training data frequency or popularity. See Tables 3 and 4 for examples of popularity statistics across basketball players and football teams.

For Figure 2 left, we use queries of the form *Tell me the year the basketball player [name] was born in* (see 1) and evaluate LLMs' accuracy. We compare the correctness of the LLM for entities (players) of varying popularity. We observe that i) LLM performance is better for entities with higher popularity, and ii) larger LLMs have better performance for entities that are less popular. Recent works show similar relationships with popular/typical input (Mallen et al., 2022; Yuksekgonul et al., 2023).

**Constrainedness of the CSP vs LLM Performance:** A well-explored complexity metric for CSPs is constrainedness (Gent et al., 1996). Here, we define constrainedness as the number of potential solutions to a given problem in the domain of the output. For instance, for a query of the form *Tell me a word that starts with the letter e and ends with the letter t*, we quantify constrainedness by the number of such words[4] in the English language that satisfy these constraints. Figure 2 right shows how constrainedness relates to success: i) LLMs have worse performance for more constrained problems, and ii) larger models perform better across all constrainedness levels.

**Summary:** We argue that the CSP lens can provide a useful view to capture the difficulty of factual queries. Our goal is to build a framework to discuss how LLMs process factual queries and generate errors. Next, we investigate LLMs' internal mechanisms to understand factual errors.

# 4 UNDERSTANDING FACTUAL ERRORS VIA ATTENTION TO CONSTRAINTS

Here, we explore how an LLM processes constraints when the model produces factually incorrect text. Geva et al. (2023); Meng et al. (2022) suggest that attention layers transfer the factual information from the source entity (e.g., *Bad Romance*) to the last token for generation (to generate *Lady Gaga*, Figure 3) when the LLM correctly addresses a query. However, these works do not explore the mechanisms when the model produces factually incorrect responses. Intuitively, we want to quantify how the LLM interacts with constraints to understand constraint satisfaction (and thus factual errors).

To study how the LLM processes a constraint, we focus on the attention *to the constraint tokens*, i.e.,

$$\boldsymbol{a}_{c,T}^{\ell,h} = A_{c,T}^{\ell,h}\big(x_c^{\ell-1} W_V^{\ell,h}\big) W_O^{\ell,h}, \tag{4}$$

---

[4]We use `nltk.corpus.words` (Bird et al., 2009) to compute the number of such words.

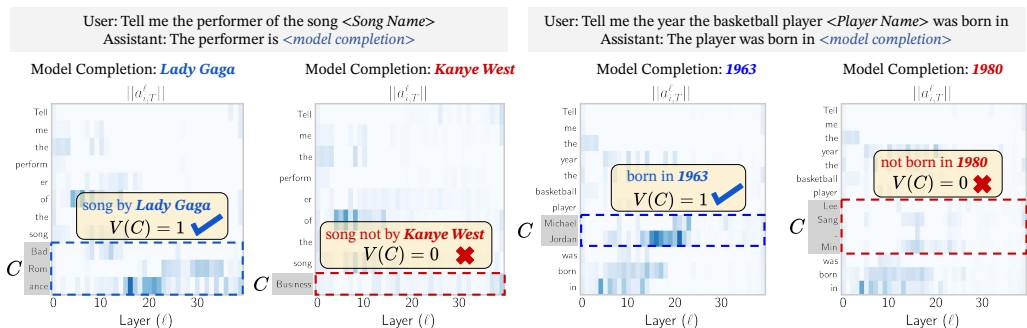

Figure 3: **Tracking attention to predict factual errors in single-constraint settings.** We track the attention contribution from the constraint tokens during generation. We observe a small-norm contribution ($||\mathbf{a}_{i,T}^{\ell}||$) when the LLM makes a factual error and a larger-norm attention contribution when the LLM is factually correct. The red text indicates factually incorrect completions, whereas the blue text indicates factually correct completions.

where $\boldsymbol{a}_{c,T}^{\ell,h} \in \mathbb{R}^d$ indicates the attention contribution from a constraint token $c$ through a head $h$ to the final token $T$ (where the $T + 1$-th token will be generated). The total contribution to $T$ is then is $\boldsymbol{a}_{c,T}^{\ell} = \sum_h \boldsymbol{a}_{c,T}^{\ell,h}$. When the constraint comprises multiple tokens denoted by the set $C$, we take the maximum value across all constraint tokens, i.e., $A_{C,T}^{\ell,h} = \max_{c \in C} A_{c,T}^{\ell,h}$ or $\mathbf{a}_{C,T}^{\ell,h} = \max_{c \in C} ||\mathbf{a}_{c,T}^{\ell,h}||$[5]. Figure 1 shows an example. We track the regions that are marked by $C_1$ and $C_2$ that represent the constraints that the movies were directed by the given directors and won the given awards.

To understand whether attention to constraints can help explain factual errors, we study three factors. First, we explore the relationship between attention and popularity of the constraining entity, as we find that LLM's correctness correlates with popularity (Fig 2). Next, we explore the relation of attention to the LLM's confidence $\hat{\mathbb{P}}(Y|X)$, which estimates the probability of a completion $Y$ given the prompt $X$. Finally, we explore how attention patterns behave when we scale the LLMs.

**Attention predicts popularity:** In Figure 11, we show the results for predicting the popularity of the constraining entity in the prompt (the basketball player) only from the attention weights ($A_{C,T}^{[L],[H]}$) using Lasso Regression. In all Llama-2 models (7B, 13B, 70B), the predicted popularities using attention values significantly correlate with the ground truth popularity (over a held-out set, with Spearman's Correlation $\rho \geq 0.65$ and p-value $p \approx 0$ for all LLMs). See Appendix C.1 for details.

Popularity seems predictive, yet we may not always have access to a clean popularity measure or frequency of constraints in training data. Our main goal is to characterize and predict factual errors, thus we seek a more reliable indicator of factual correctness.

**Attention correlates with confidence and LLM's correctness:** In Figure 4 (left four panels), each row represents the attention flow across layers for a single sample and we sort the points by the confidence of the LLM. The leftmost panels show the attention for the 25 most confident predictions and the middle panels show the 25 least confident predictions, where the x-axis shows the layers, and colors indicate the norm of the attention contribution from the constraints ($||\boldsymbol{a}_{C,T}^{\ell,[H]}||$). The core observation is that *when the LLM is accurate, there is more attention to constraint tokens* (first column) in sharp contrast to cases where the LLM fails and the attention is weak (second column).

In Figure 4's rightmost plots, queries are sorted and grouped by the LLM's total attention contribution from the constraints across all layers ($\sum_{\ell} ||\mathbf{a}_{C,T}^{\ell}||$), and LLM's accuracy is computed for each group. We observe that *the magnitude of attention to constraints correlates with accuracy*. This observation is not only interesting in hindsight; aforethought could have suggested either outcome (e.g., more attention correlating with hallucination). This is a positive observation that suggests attention to constraints can be used to predict LLM's success.

**Language models grow larger, pay more attention, and succeed more:** In Figure 5, each panel compares the attention to constraints for the basketball player queries between two different LLMs,

---

[5]While there is earlier work that suggests the last constraint token could the most important, we observed that in practice there are subtleties. See Appendix C.2 for a discussion.

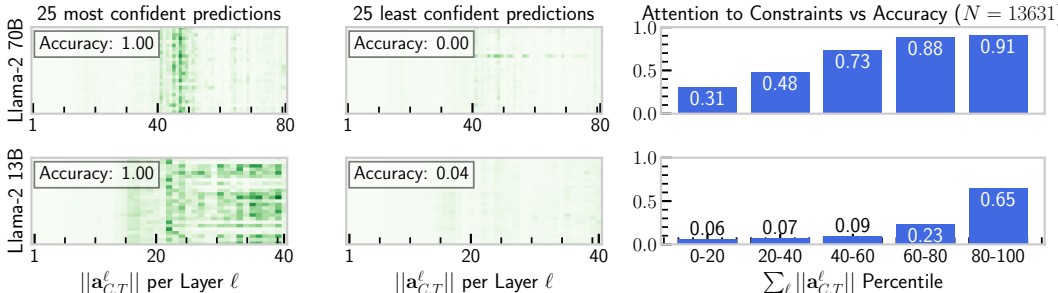

Figure 4: **Attention correlates with correctness. The first two columns of panels** give the 25 samples for which the LLM makes the most and the least confident predictions. The color indicates the norm of the attention contribution from the constraint, where each column in the panel captures a layer in the LLM and each row is a specific sample. **The last column of panels** relates the total attention to constraints and accuracy, where the x-axis is the attention contribution percentile in the dataset and the y-axis is the accuracy in the bin. The results are for the year of birth queries (see 16).

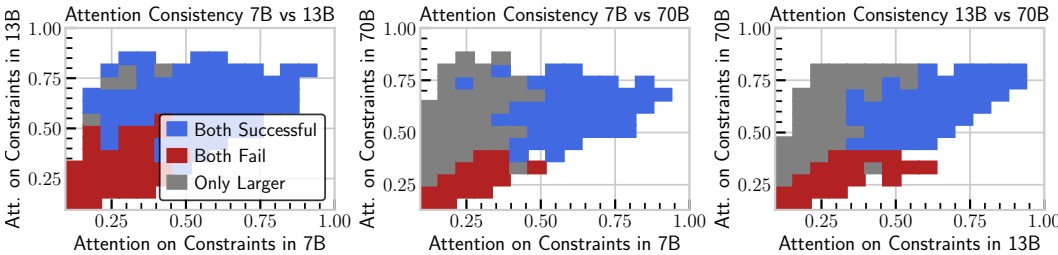

Figure 5: **Attention contribution and model scaling.** Here, the x-axis and y-axis show the attention to the constraints for the smaller LLM and the larger LLM, respectively, and are normalized by dividing by the maximum value of total attention in the dataset. Coloring is determined by which of the two LLMs succeeds in factual queries. We group the factual queries by their x-axis value and y-axis values and color the cell with the most frequent category in the cell. Appendix Figure 13 presents the complete scatter plot.

where the x-axis indicates the smaller LLM, the y-axis indicates the larger LLM and the coloring indicates the success of the pair of LLMs. We group prompts by the attention contribution, and color the cells by the the most frequent category. We find that more (relatively) attention in both LLMs generally indicates success for both and less attention in both LLMs indicates failure for both. For cases on the top left, the larger LLM does pay more attention, and only the larger LLM succeeds. Overall, we note a consistent pattern between attention and correctness across model scales; performance improvements in larger LLMs correlate with increased attention to constraint tokens.

**Summary:** We explored the interaction between attention, constraints, and factual correctness. Our analyses suggest that attention can help us understand and predict factual errors. Next, we further pull this thread with extensive experiments to predict factual errors using LLMs' attention patterns.

## 5 PREDICTING FACTUAL ERRORS USING ATTENTION TO CONSTRAINTS

We now show how our mechanistic understanding can be used to predict the failures of LLMs. Let $X$ denote a prompt: a sequence of tokens that specifies a factual query with a set of constraints $\mathcal{C} = \{(C_1, V_1), \ldots (C_K, V_K)\}$. Let $\hat{Y}$ be the response tokens obtained from the LLM after feeding $X$. Broadly, we want to design a function $f$ to estimate the probability that a constraint $k$ is satisfied:

$$\hat{\mathbb{P}}(V_k(\hat{Y}) = 1) = f(X, \hat{Y}, C_k, \mathcal{M}),$$

using the prompt, completion, constraint tokens, and the LLM $\mathcal{M}$. For single-constraint factual queries where there is a single correct completion $Y$, this can be reduced to the correctness $\hat{\mathbb{P}}(Y = \hat{Y}) = f(X, \hat{Y}, \mathcal{M})$. Note how this formalism closely matches that of selective classification (Geifman & El-Yaniv, 2017), where the goal is to abstain when the model would otherwise fail.

| Dataset Name | Constraint Type(s) | $N$ | Constraint Source | Verifier | Example Prompt |
|---|---|---|---|---|---|
| Basketball Players | *born in the year* | 13631 | WikiData | Exact Match | Figure 16 |
| Football Teams | *founded in the year* | 8825 | WikiData | Exact Match | Figure 17 |
| Movies | *directed by* | 12197 | WikiData | Exact Match | Figure 18 |
| Songs | *performed by* | 2813 | WikiData | Exact Match | Figure 19 |
| CounterFact | *mother tongue* | 919 | CounterFact | Exact Match | Figure 20 |
| CounterFact | *citizenship* | 958 | CounterFact | Exact Match | Figure 21 |
| CounterFact | *headquarter location* | 756 | CounterFact | Exact Match | Figure 22 |
| Books | *author, published year* | 1492 | WikiData | WikiData Search | Figure 23 |
| Nobel Winner | *won Nobel, born in city* | 1290 | Opendatasoft (2023) | WikiData Search | Figure 24 |
| Words | *starts with, ends with* | 1352 | Hand-Curation | Character Match | Figure 25 |

Table 1: **Overview of Datasets.** The columns denote the dataset name, constraint type, number of prompts, and the data sources used to collect entities and verify LLM responses, respectively.

**Datasets:** For our evaluations, we curate a benchmark with 10 datasets that are listed in Table 1 containing over $40,000$ queries. For single-constraint queries, we curate 4 datasets using WikiData and 3 datasets using the existing CounterFact dataset (Meng et al., 2022). We further designed three 2-constraint datasets, using WikiData (Books), Opendatasoft (2023) (Nobel Winners), or hand-curation (Words). Further details about all data curation can be found in Appendix D.

**Constraint Verification** $(V_k)$**:** We use Exact Match[6] for single-constraint queries with a single solution. We probe WikiData to verify constraints when queries have multiple potential solutions. Appendix D.3 contains a complete description of the methodology.

**Models:** We use the 7B, 13B, and 70B parameter variants of Llama-2 (Touvron et al., 2023) released at HuggingFace's Transformers (Wolf et al., 2019). We perform our experiments on a single NVIDIA A100-PCIE-80GB GPU. 80GB memory can only fit the Llama-2 70B in 8-bit precision (Dettmers et al. (2022a) report marginal-to-no performance drop). See Appendix A for further details on models.

**Evaluation Metrics**: We use AUROC for the binary task of predicting failure or success as it does not require setting a threshold for the classifier. We also report $\text{Risk}_{\text{Top 20\%}}$ (the fraction of errors for the samples with top 20% of the scores by the predictor $f$), $\text{Risk}_{\text{Bottom 20\%}}$ (the fraction of errors for the samples with the bottom 20% of the scores by the predictor $f$). These metrics measure how well the model performs on the most and least reliable completions according to the predictor $f$. For a good success predictor, we want the error fraction to be low among high-score examples (small $\text{Risk}_{\text{Top 20\%}}$) and have a large fraction of failures among low-score examples (large $\text{Risk}_{\text{Bottom 20\%}}$).

## 5.1 PREDICTING FACTUAL CORRECTNESS

**Predictors** $(f)$**:** We propose the constraint satisfaction probe, SAT PROBE, that predicts if an individual constraint is satisfied by only looking at self-attention. To demonstrate the simplicity, we let $f$ be a linear function of the attention weights to constraints:

$$\hat{\mathbb{P}}(V_k(\hat{Y}) = 1; A_{C_k,T}) = \sigma(w^T A_{C_k,T} + b),$$

where $A_{C_k,T}, w \in \mathbb{R}^{L \times H}, b \in \mathbb{R}, A_{C_k,T} = \{\forall \ell \in [L], h \in [H] : A^{\ell,h}_{C_k,T}\}$. We linearly probe the attention weights across all layers and attention heads and estimate $w$ and $b$ using Logistic Regression. In multi-constraint settings we use SAT PROBE and combine the predictions for constraints:

$$\hat{\mathbb{P}}(\prod_{k \in [K]} \mathbf{1}_{\{V_k(\hat{Y})=1\}}; A_{C_k,T}) = \prod_{k \in [K]} \hat{\mathbb{P}}(V_k(\hat{Y}) = 1; A_{C_k,T}).$$

**Baselines:** We compare SAT PROBE to the CONFIDENCE of the model, $\hat{\mathbb{P}}(\hat{Y}|X)$, which concurrent work reports as a hallucination detector (Varshney et al., 2023); and a CONSTANT predictor that predicts the majority class (either 0 or 1) as baselines. Note that while CONFIDENCE is a strong baseline, it only provides an overall estimate for the whole generation, and cannot predict the failure

---

[6]We acknowledge that exact match is a strict criterion that could introduce noise to our evaluations, and it constitutes a limitation where we use this verification. Evaluating factual correctness is still an evolving research topic (Min et al., 2023) and we do our best to find queries and prompt structures that suffer the least from this.

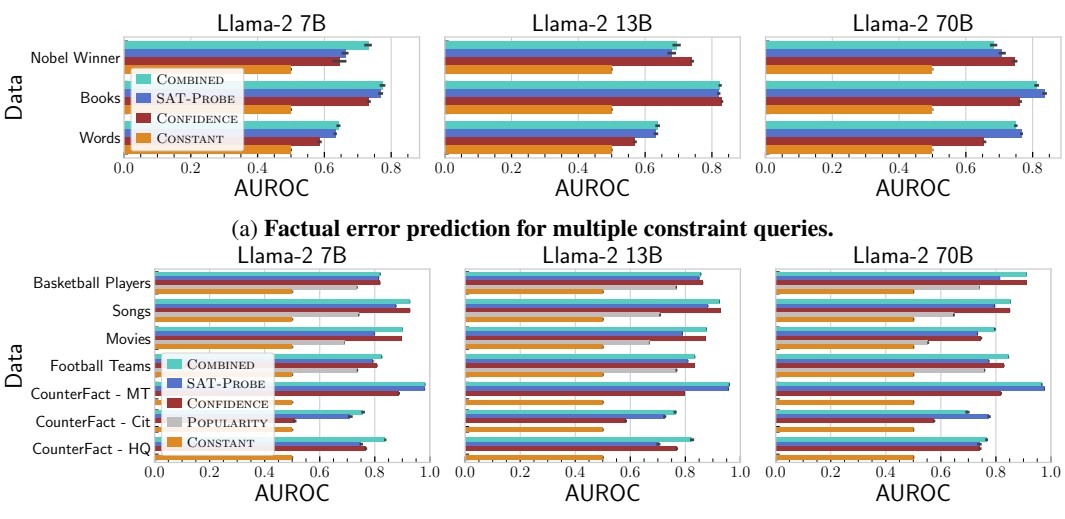

(a) **Factual error prediction for multiple constraint queries.**

(b) **Factual error prediction for single constraint queries.**

Figure 6: **Factual Error Prediction.** SAT PROBE is often comparable and marginally worse than CONFIDENCE, and better than POPULARITY. COMBINED predictor where attention and confidence are combined to predict failures mostly performs the best. Error bars show the standard error over 10 different random train/test splits. Tables 5,6 contain the results in tabular form.

for individual constraints. We also use the POPULARITY baseline only in the single-constraint WikiData datasets that we curated – as in other datasets, it is not accessible (CounterFact) or unclear how to compute (multi-constraint). In the Appendix, we also give results with featurization using the attention contribution, e.g., $\mathbf{a}_{C_k,T} = \{\forall \ell \in [L], h \in [H] : ||\mathbf{a}_{C_k,T}^{\ell,h}||\}$, denoted by SAT PROBE(a). Finally, when predicting constraint satisfaction, we concatenate confidence to the set of attention features to the logistic regression and denote this by COMBINED.

**Results:** In Figure 6a, we present the overall AUROC of predicting factual correctness for multi-constraint queries, and Table 6 contains all metrics. In this task, we find that SAT PROBE performs comparably to the model's CONFIDENCE in the correctness prediction task, in addition to being able to provide fine-grained feedback (i.e., which constraint is not satisfied, see §5.2). Figure 6b presents the AUROC results for the single-constraint setting. In the single-constraint setting, SAT PROBE is comparable to CONFIDENCE. Further, we find that the approaches are comparably good in isolating highly reliable vs unreliable points (Table 5,6 show Risk$_{\text{Top 20\%}}$ and Risk$_{\text{Bottom 20\%}}$ metrics). CONFIDENCE alone appears to be slightly more performant than attention alone, yet attention helps us more efficiently and identify the failures in a fine-grained manner (see § 5.2). COMBINED predictor that uses both attention and confidence gets the best performance across all three metrics in most scenarios. In Figure 15, we compare the predictions made by SAT PROBE and CONFIDENCE. Notably, while they correlate, both predictors have different errors, e.g., there are cases where the LLM is overconfident yet there is almost no attention and it is a factual error, or vice-versa. This further provides insight into why the COMBINED predictor performs better overall.

Attention alone is significantly better than the CONSTANT baseline which suggests that the signals relay a nontrivial amount of information, sometimes exceeding CONFIDENCE. Surprisingly, even though LLMs are optimized by maximizing the next token probability, probing attention patterns exclusively on the constraints can match or sometimes exceed this performance—*without using hidden states or non-constraint tokens*. However, attention alone does not explain all failures (we observe some attention on constraints where the model still fails), there is an opportunity for further investigation. *Our findings demonstrate the value of studying the procedure by which a model produces an output, rather than only the output itself.*

## 5.2 EXTENSIONS

We study 3 extensions to explore the potential of SAT PROBE and propose avenues for future work.
**Early stopping:** Using SAT PROBE, we can predict failures partway through the computation and

save costs. Appendix Figures 7 shows that we can predict failures earlier in the inference with an experiment across all single-constraint datasets. Specifically, we use only attention weights up to an intermediate layer and predict failures ahead of time. For Llama-2 7B and 13B, we can stop the inference early without degradation in the average performance and save 50% of wall-clock time on failures for most datasets. In 70-B, early stopping results in a slight drop in performance. For use cases where we have a high $\text{Risk}_{\text{Bottom 20\%}}$, we can isolate these most unreliable predictions and abstain from making a prediction. See Appendix B.4 for details on the ablation.

**Predicting partial constraint satisfaction:** SAT PROBE gives access to failure predictions for individual constraints. We report the partial constraint satisfaction results in Table 7 where we report the failure prediction metrics for individual constraints and find comparable results to the single-constraint prediction task. While SAT PROBE lets us test whether each constraint is satisfied, using the raw CONFIDENCE does not since it only outputs a single value for all constraints. We believe producing fine-grained reliability statements, such as reporting partial constraint satisfaction, can prove useful for debugging (e.g., failing to follow specific instructions).

**Generalized predictors:** We explore using a single failure predictor across all constraint types. For this purpose, we train a failure predictor on a mixture of single constraint datasets and report the performance over individual datasets in Appendix B.5 and Figure 8. We observe the performance is competitive with training individual predictors for each constraint and better than POPULARITY. This suggests the potential of pursuing efforts to construct general factual error detectors.

## 6   RELATED WORK

Carlini et al. (2021; 2022); Biderman et al. (2023) related the training data frequency of a string to memorization in LLMs. In recent work, Mallen et al. (2022); Kandpal et al. (2023); Sun et al. (2023) document the relation between the success/difficulty of factual queries and a measure/proxy for training data frequency. Several recent works investigated the mechanics of factual recall. Numerous works Elhage et al. (2021); Devlin et al. (2018); Olsson et al. (2022); Clark et al. (2019); Tian et al. (2023); Htut et al. (2019); Voita et al. (2019); Burns et al. (2022); Gurnee et al. (2023) analyzed how specific attention heads exhibit certain functionalities, such as heads that encode syntax or induction heads that copy tokens. Further, Meng et al. (2022); Geva et al. (2023) discuss the role of attention in specifically transferring factual information, and Hernandez et al. (2023) studies how specific relations can be decoded with a linear transformation from the subject tokens. However, these works do not investigate the generation of factually incorrect text. Halawi et al. (2022); Belrose et al. (2023) study how LLMs internally deal with safety-critical input, such as false demonstrations for in-context learning or prompt injections. Varshney et al. (2023) detect and mitigate hallucinations using the model's CONFIDENCE. Closest in spirit, in concurrent work Li et al. (2023) seek directions that encode 'truthfulness' in the total attention contributions across all tokens and use these to intervene on LLMs; whereas we focus specifically on predicting factual errors, showing one can use only attention weights and look only at the constraints, contribute the analyses around fine-grained constraint satisfaction and early stopping. Mündler et al. (2023); Manakul et al. (2023); Zhang et al. (2023) interact with the LLMs in a black box fashion and aim to determine factual errors through inconsistencies, but doing so requires several forward passes and conveys conflicting signals such as refuting an initial claim, which can diminish user trust (Liao & Vaughan, 2023; Huang et al., 2020).

## 7   CONCLUSION AND FUTURE WORK

Our analysis provides initial insights into leveraging LLM's internals to understand factual errors and it raises several exciting questions for future work. We studied only conjunctive factual queries, but the class of potential constraints is much broader (e.g. instructions (Ouyang et al., 2022), disjunctive queries). While this framework extends the scope of mechanistic investigations of factuality in LLMs, it does not cover all possible factual queries to LLMs. In addition, this work assumed access to the constraint tokens; moving forward it is interesting to consider cases where we need to extract this information. The content of the information in attention patterns remains opaque and warrants further investigation. Future work could investigate how to perform different actions to fix errors, such as steering the model behavior by manipulating the attention to constraints. We hope that the methods and analyses of our study will help with the pursuit of reliable and safe generations from LLMs.

ACKNOWLEDGMENT

We would like to thank Duygu Yilmaz, Marah Abdin, Rahee Ghosh Peshawaria, Ekin Akyurek, Federico Bianchi, Kyle Swanson, Shirley Wu, James Zou, Eric Horvitz, Zhi Huang, Marco Tulio Ribeiro, Scott Lundberg, Dilara Soylu for their support and comments throughout the project.

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

## A    MODEL DETAILS

We use the Llama-2 family models released in Touvron et al. (2023) through the HuggingFace Transformers library (Wolf et al., 2019). To fit Llama-2 70B in a single A100 GPU with 80GBs of memory, we use 8-bit quantization with `bitsandbytes` (Dettmers et al., 2022b;a). For all of the models, we sample from the model using greedy decoding and temperature 0.

| Model Name | Model ID on HuggingFace Transformers | $L$ | $H$ |
|---|---|---|---|
| Llama-2 7B | `meta-llama/Llama-2-7b-hf` | 32 | 32 |
| Llama-2 13B | `meta-llama/Llama-2-13b-hf` | 40 | 40 |
| Llama-2 70B | `meta-llama/Llama-2-70b-hf` | 80 | 64 |

Table 2: Overview of Models.

Touvron et al. (2023) reports that since key-value caches become prohibitive for larger models such as the Llama-2 70B, they use Grouped-Query Attention (GQA, (Ainslie et al., 2023)) (as opposed to Multi-Head Attention) with 8 key-value projection variant. GQA divides each query head into multiple groups, which reduces the size of the KV cache to be loaded into the memory at each inference step. However, this does not change our analyses which focus on the attention weights ($A$) or the attention contribution ($a$).

The *MLP contribution* in the Llama-2 family is computed by

$$\mathbf{m}_i^\ell = W_F^\ell \, \sigma(W_I^\ell(\mathbf{a}_i^\ell + \mathbf{x}_i^{\ell-1})) \tag{5}$$

where $W_F^\ell \in \mathbb{R}^{d \times d}$, where $W_I^\ell \in \mathbb{R}^{d \times d}$. Unlike attention, the input to the MLP updates for a token only uses the representation of the *same* token. Since we are interested in the interaction between the generation and the constraint tokens, we explore only the attention contribution in this work. It is possible that the MLP contribution has relevant signals, and we leave this exploration to future work.

## B    EXPERIMENTAL RESULTS - EXPANDED

### B.1    SINGLE-CONSTRAINT EXPERIMENTS

In the single-constraint experiments, we evaluate the methods over the datasets and constraints documented in Table 5. For each dataset, we split the dataset into two sets (train and test) and normalize each feature to zero mean and unit variance using the training split. We train a Logistic Regressor with $C = 0.05$ $L_1$ regularization on one subset and evaluate the performance on the other subset. We repeat this experiment with 10 random seeds and report the mean performance and the standard error next to it in Table 5 for each dataset and method. In the Tables, SAT PROBE(A) indicates using attention weights as the input to the probe, whereas SAT PROBE(a) indicates using the attention contribution as the probe. The latter is closer in spirit to Li et al. (2023) that explores the entire attention contribution through an individual head ($\mathbf{a}_i^{\ell,h}$) whereas we focus on contributions from only the constraint tokens. In addition, we only look at the attention weights with the former probe, which performs better than SAT PROBE(a).

### B.2    MULTI-CONSTRAINT EXPERIMENTS

In the multi-constraint experiments, we evaluate the methods over the datasets and constraints documented in Table 6. We follow the similar training/test split protocol as in §B.1. Additionally, when doing the train/test splits, we do not leak the same pair of constraints from the test set to the training test. Specifically, since for a pair of constraints we can have two prompts (e.g. *starts with e and ends with r* vs *ends with r, starts with e*), we split samples based on the set of constraints in the prompts. In this case, factual correctness is defined as satisfying both of the constraints in the query. We provide the results in tabular form in 6.

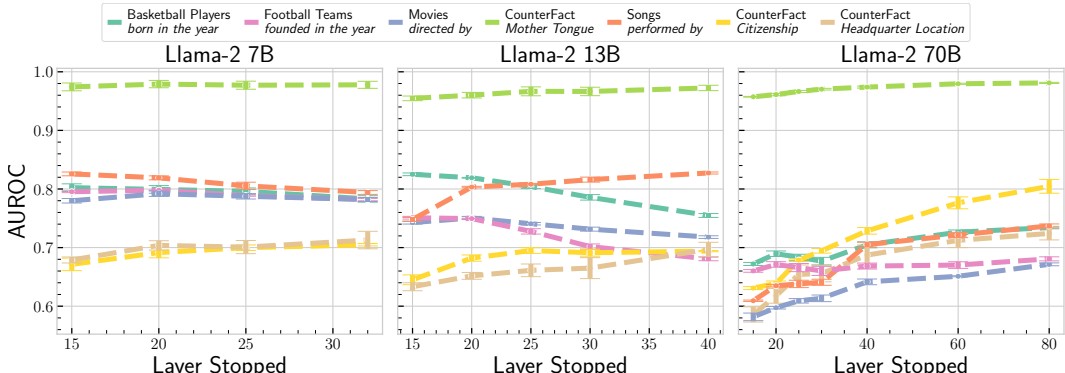

Figure 7: **Early Stopping Analysis for Inference, decomposed per dataset.** Here, the x-axis indicates the layer until which we use the attention weights as the failure predictor ($A_{C,T}^{[L],[H]}$), the y-axis indicates the average performance across 3 random seeds for the single constraint datasets, and the color indicates dataset. Overall, for Llama-2 7B and 13B, we observe that we can stop the inference early in the process without degradation in the average performance and save 50% of wall-clock time for most datasets.

## B.3 PREDICTING PARTIAL CONSTRAINT SATISFACTION

Here, we follow the same protocol as in §B.1, however, we predict constraint satisfaction for individual constraints in multi-constraint queries. Note how CONFIDENCE cannot be a baseline here – it simply produces a probability that a completion follows a prompt, and cannot make fine-grained statements about individual constraints. We believe future work could pull this thread to reason about individual instructions, factual queries with different compositions (e.g. logical operators such as OR), and beyond.

## B.4 EARLY STOPPING EXPERIMENTS

Here, we explore whether we can predict failure ahead of time to stop the execution and save computing time. Specifically, we use the attention weights up to a layer, e.g. $A_{C,T}^{[L'],[H]}$ for $L' < L$. We repeat this experiment across multiple choices of layers with three random seeds, and report the results in Figure 7. Overall, for most datasets, there is little to no performance drop in terms of failure prediction, even if we use up to 50% less computation time. This is less of the case for Llama-2 70B, where using later self-attention layers provides more value to the performance. We believe using the model's attention patterns could potentially be used to predict failure ahead of time, potentially abstaining from making a prediction or escalating the request to potentially a larger model e.g. similar to a cascading setting (Wang et al., 2011).

## B.5 GENERALIZING THE FAILURE PREDICTOR

Here, we explore whether we can train a single failure predictor across multiple constraints. To do so, we create a mixture of datasets by mixing data from each of the relations. Specifically, for each of the single-constraint datasets in Table 1, we split 50% of the dataset for training, and leave 50% of the dataset for testing. We train a Logistic Regressor on the training set and quantify the performance in the test set. We perform this with 5 random seeds, where the randomness is over the train/test splits. In Figure 8 and Table 8, we present the results. In particular, when compared to training individual predictors as in Figure 6b, we observe minor performance fluctuations. We believe this suggests the potential of training general-purpose failure predictors using attention maps. We leave a detailed exploration of this angle for future work.

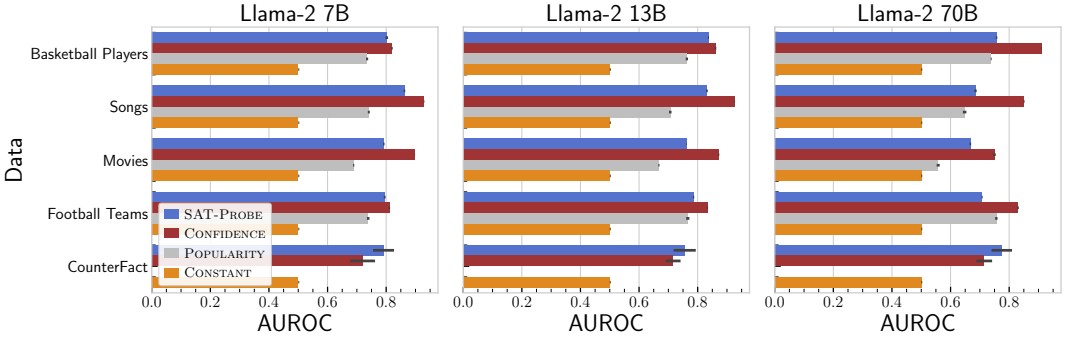

Figure 8: **Single Predictor for All Constraints.** We train a single failure predictor on a mixture of datasets and test the performance across held-out samples of the same datasets. Overall, the performance remains better than POPULARITY and competitive with training individual predictors, which is reported in Figure 6b.

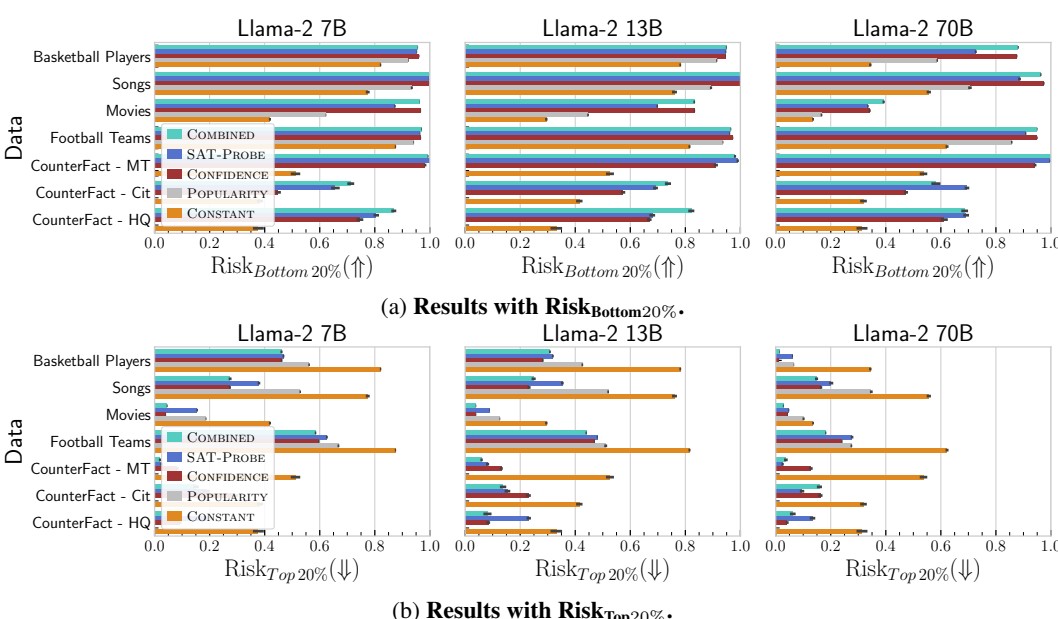

Figure 9: **Factual Error Prediction.** (a) Single-constraint failure prediction results for the Risk$_{Bottom20\%}$ metric. (b) Single-constraint failure prediction results for the Risk$_{Top20\%}$ metric.

## C  DETAILS ON THE EXPLORATORY ANALYSES

### C.1  PREDICTING POPULARITY

For the basketball players dataset, we filter the dataset for popularity values $\geq 15$, since in the lower end we find that there is less meaningful signal for comparison. After this filtering, we end up with 4022 entities. We split these into two sets of equal size (training and test), and we perform Lasso regression in scikit-learn (Pedregosa et al., 2011) on the training set with 0.005 regularization strength to regress the popularity values. We then use the regressor to make predictions over the held-out set and compute the Spearman rank-order correlation coefficient computed using scipy.stats.spearmanr (Virtanen et al., 2020) and the p-value for testing the null hypothesis that there is no ordinal correlation.

### C.2  COMPUTING ATTENTION TO A CONSTRAINT

**Taking the Maximum over Constraint Tokens:**  Previous works report that the last subject tokens have the largest significance in terms of tracing the information (Meng et al., 2022). While we

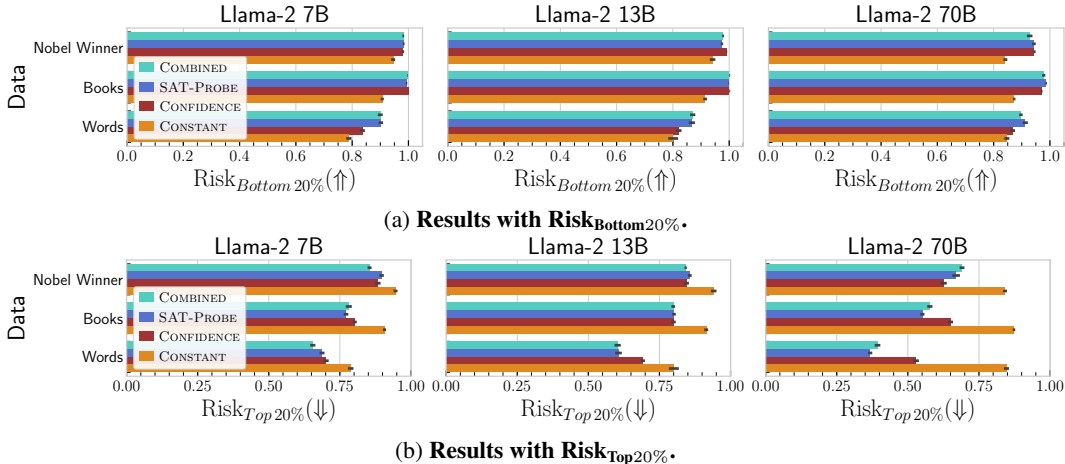

(a) **Results with RiskBottom20%.**

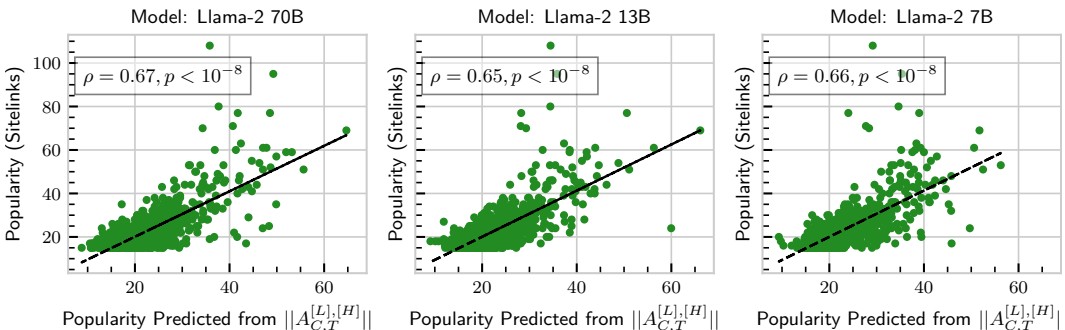

(b) **Results with RiskTop20%.**

Figure 10: **Factual Error Prediction.** (a) Multi-constraint failure prediction results for the $\text{Risk}_{\text{Bottom20\%}}$ metric. (b) Multi-constraint failure prediction results for the $\text{Risk}_{\text{Top20\%}}$ metric.

Figure 11: **Predicting constraining entity popularity via attention.** We report the results for the basketball players dataset. The x-axis indicates the popularity values predicted using attention weights and lasso regression, the y-axis indicates the ground truth popularity values, and each column belongs to a model (indicated in the title). Boxes on the top left contain the Spearman correlation and p-value for testing the null hypothesis that there is no ordinal correlation.

observe this to be generally the case, there are subtleties that arise for end-to-end evaluations. For instance, `Kelly Oubre Jr.`, we observe very little attention to `Jr.` and the most attention is on `Oubre`. In Figure 12 we have two example prompts. Overall, due to the subtleties in tokenization and potential ambiguity in the definition of what is the core part of a constraint, we choose to take the `max` over all constraint tokens, instead of only looking at the last token.

# D DATASETS

## D.1 DATA CURATION FOR SINGLE-CONSTRAINT QUERIES

**WikiData:** For 4 groups of queries that are not from CounterFact, we probe WikiData[7]. For instance, for basketball players, we searched WikiData for all basketball players with at least 5 site links on the page where this criterion was done to ensure quality (entities with fewer site links did not always turn out to be professional basketball players). This criteria also avoids sampling from extremely tail knowledge, since as demonstrated in Sun et al. (2023), this can be problematic for model performance and we are seeking settings where the model can perform reasonably well, albeit not perfectly. Next, we retrieve the relevant field (e.g. in this case year of birth) of the entity. The choice of relevant fields for constraints was also guided by i) the need to avoid extreme tail knowledge, and ii) the feasibility of the Exact Match metric. For instance, the year of birth or year of founding can simply be evaluated

---

[7] https://query.wikidata.org/sparql

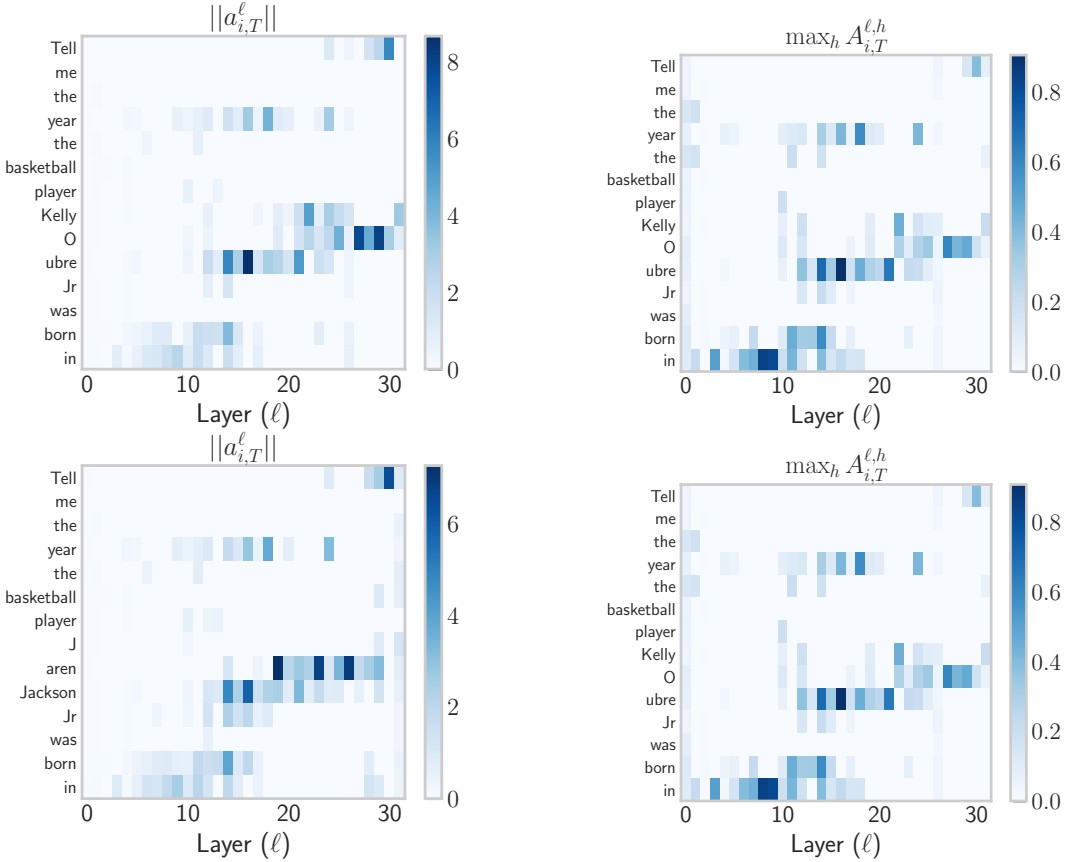

Figure 12: **Attention for players that have 'Jr' in the name.** Overall, while we frequently observe that there is large attention on the last token of the constraining entity, in practice there are many cases in which earlier tokens are more informative. In the above examples, we look at the attention contributions (left column) and the attention weights (right column) for Llama-2 7B with the basketball player prompts, e.g. Fig. 16.

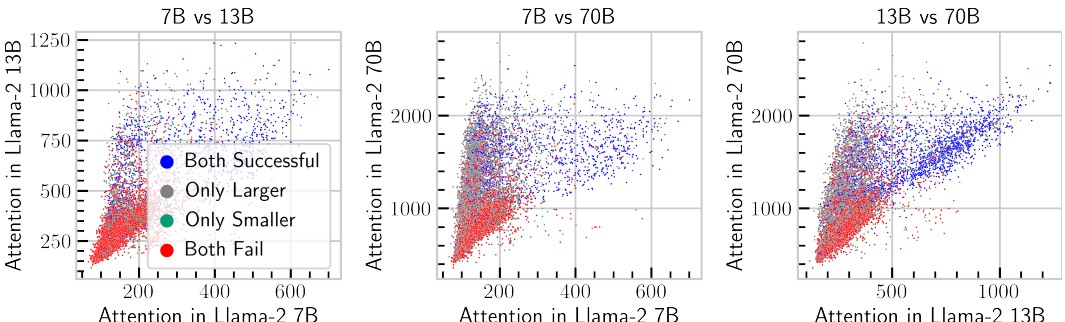

Figure 13: **Attention Mass and Model Scaling. Scatterplot version of Figure 5.**

by the exact match between the correct year and the year the model produces (e.g. see queries 16,17). We chose the prompting strategy such that it is well set up for the model and we can expect that the token that follows will likely contain important factual information.

**CounterFact** (Meng et al., 2022)**:** We picked the three relations in Table 1 from CounterFact as they were arguably the least vulnerable to the strictness of the exact match evaluation. Specifically, we found that there is less variation in terms of potential correct responses for mother tongue queries (e.g. Figure 20), citizenship queries (e.g. Figure 21), or headquarter location queries (Figure 22). This is arguably visible from the model performance in Table 5 where we see nontrivial model performance albeit we use Exact Match as our evaluation metric.

## D.2 DATA CURATION FOR MULTI-CONSTRAINT QUERIES

We curated three multi-constraint datasets where it is relatively easy to perform constraint verification to label model completions for factual correctness. All of these datasets have at least one correct completion for each query.

**Words:** For the word queries, we iterate over letters of the alphabet, and generate prompts that contain *starts with* and *ends with* constraints for all pairs of letters, resulting in $26 \times 26 = 676$ pairs of letters in total, and upon using the permutations of constraints, we in total have 1352 prompts.

**Nobel Winner:** We use the Nobel Prize Winners dataset (Opendatasoft, 2023) to generate queries with at least one potentially correct completion. For each unique city of birth in this dataset (645), we generate one prompt using the template in Figure 24, asking for the name of a Nobel Prize Winner who was born in a given city. Upon using the permutations, we have a resulting set of 1290 prompts.

**Books:** We probe WikiData to collect a dataset of authors with books. From WikiData, we scrape a collection of 746 books, with a pair of authors and years of publication. For each pair, we generate a query, and using the permutation of constraints, we end up with 1492 prompts.

## D.3 VERIFICATION

**Exact Match:** Even though the Exact Match criterion is fairly strict, it is still a common metric used in various evaluations of LLMs from classification to question answering(e.g. Liang et al. (2022)). Let $Y$ be the ground truth completion, and let $\hat{Y}$ be the model's completion, where both are sequences of tokens. We simply check whether the first $|Y|$ tokens of the $\hat{Y}$ sequence are the same as $Y$, and return $V(\hat{Y}) = 1$ if that is the case. Otherwise, we return $V(\hat{Y}) = 0$. To mitigate how strict the criterion is, we tried to set the prompts up such that it is more likely that the model completes in the desired format (e.g. see Figure 17 where we include 'the year founded in' in the question, and the assistant's response starts with 'The team was founded in').

**WikiData Probing:** For multi-constraint queries, the evaluation is more challenging. For instance, in a query where we ask the model to return a book *whose author is* $C_1$ and the book *was published between* $C_2$, we would like to label whether indeed there was such a book from that author, and separately whether there was a book published in that year. To do so, we perform the following steps:

1. First, obtain the model's response up to any new line token (\n), if present.
2. We use this entity to search WikiData.
3. If there was no such entity (e.g. book), we return $V_1(\hat{Y}) = 0$, $V_2(\hat{Y}) = 0$.
4. If we find such an entity (e.g. book), we individually go through the relevant fields. For instance, we retrieve the list of authors and the year the book was published.
5. Once the list is retrieved, for each constraint and list, we probe the `gpt-3.5-turbo` API with the prompt in Fig 14 whether the item is in the list. If the items are found, we return 1.

We use GPT 3.5 (`gpt-3.5-turbo` endpoint) in the loop since Exact Match with e.g. arbitrary book names is a lot trickier than what we have in the single constraint queries (e.g. verifying the year of birth is easy with Exact Match). Since this is not a cheap step, we use this strategy only for multi-constraint queries.

**Character Match:** For the Words dataset, the constraints are simply of the form *starts with the letter* or *ends with the letter*. This is simple to verify, as it is simply checking the first character or the last character of the first word in the completion.

## E COMPARING CONFIDENCE AND ATTENTION

In Figure 15, we investigate whether attention and confidence behave differently when predicting factual errors. Specifically, for the Football Teams and Basketball Players datasets, we make predictions on the test sets of the datasets, and plot the predicted failure probability by the attention (y-axis), and by the confidence (x-axis). We color the bins indicating which of the models make a correct prediction (assuming a decision threshold of $0.5$).

**SYSTEM PROMPT**: You are an AI assistant that helps the user verify whether an entity is captured in a list.
**PROMPT**: I will give you one entity and one list, and I want you to respond with "YES" if the entity is within the list, "NO" if it is not in the list.
List: {List of Items}
Entity: {Query Entity}
Give your response in the following format:
'Reference in the list: item in the list if exists, None otherwise
Answer: YES or NO' and say nothing else.

Figure 14: **GPT3.5 command used for constraint verification**. Once the relevant lists are obtained from WikiData, we probe GPT3.5 for verifying whether the constraining entity is in the list we retrieved from WikiData.

| Popularity | Sampled Players |
|---|---|
| 0-10 | Gur Shelef, Kimmo Muurinen |
| 10-20 | Kevin Willis, Krešimir Lončar |
| 20-30 | Bogdan Bogdanović, Candace Parker |
| 30-40 | Joakim Noah, Miloš Teodosić |
| 40-50 | John Stockton, Luka Dončić |
| 50-60 | Dennis Rodman, Yao Ming |
| 60-70 | Kevin Durant, Stephen Curry |
| 70-80 | Magic Johnson, Larry Bird |
| 80-90 | R. Kelly, Kareem Abdul-Jabbar |
| 90-140 | Michael Jordan, Kobe Bryant |

Table 3: **Basketball Players in Each Popularity Interval.** Popularity indicates the number of site links on the WikiData page of the player.

We observe that attention and confidence do not always agree in predicting failures. In particular, there are cases where the confidence-based predictor suggests a factual error and attention-based does not, and it is not a factual error (see left); or vice-versa.

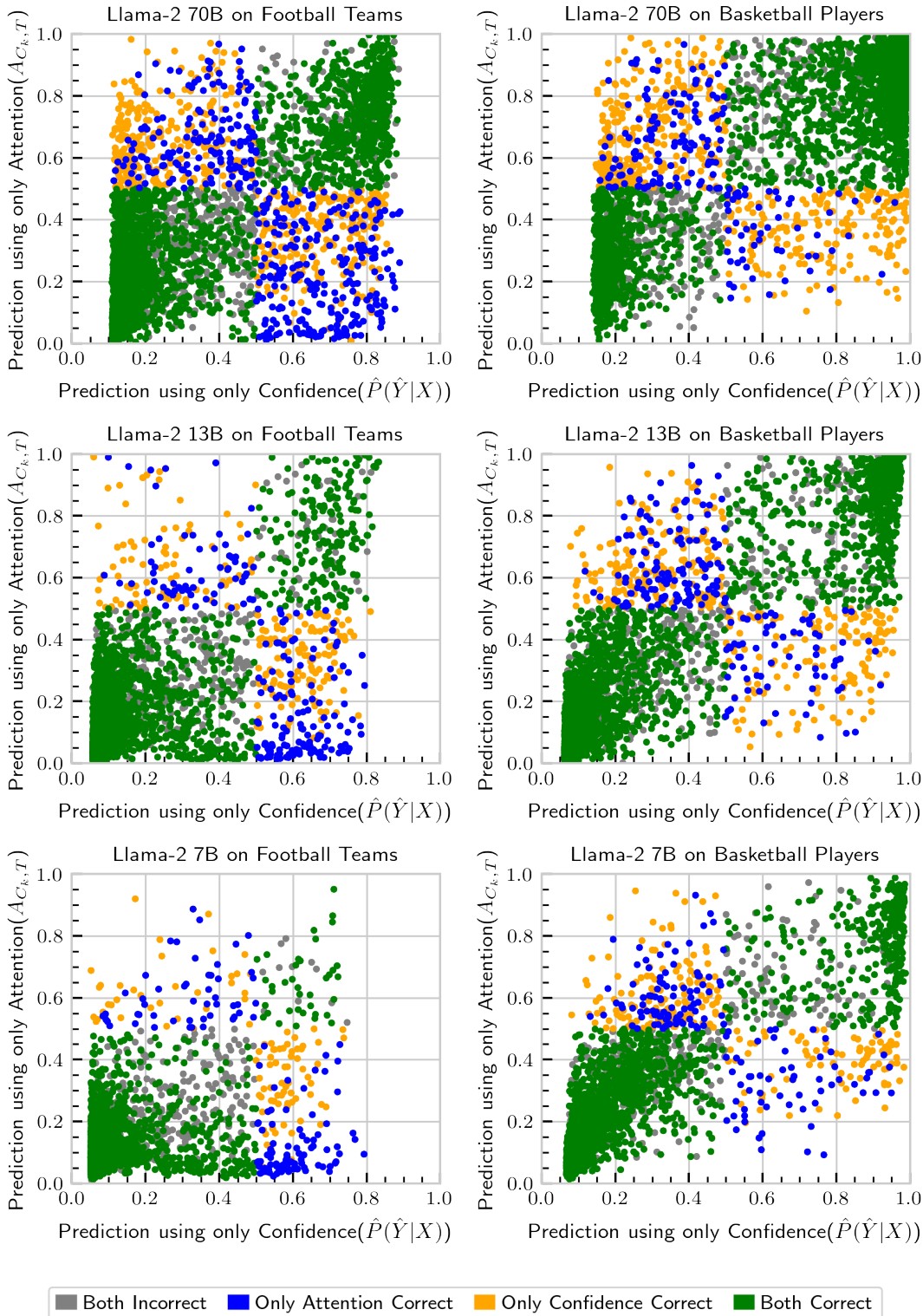

Figure 15: **Comparing Attention- and Confidence-based Predictions.** Each row indicates samples from a model. The x-axes indicate the prediction made using only confidence, y-axes indicate the prediction made using only attention. Overall, predictors do not always agree, and the disagreement becomes more significant as the LLMs are larger.

User: Tell me the year the basketball player Michael Jordan was born in.
Assistant: The player was born in

Figure 16: Example Prompt for the Basketball Players (*born in the year*) dataset queries.

User: Tell me the year the football team FC Barcelona was founded in.
Assistant: The team was founded in

Figure 17: Example Prompt for the Football Teams (*founded in the year*) dataset queries.

User: Tell me the director of the movie Titanic.
Assistant: The director is

Figure 18: Example Prompt for the Movies (*directed by*) dataset queries.

User: Tell me the performer of the song Bad Romance
Assistant: The performer is

Figure 19: Example Prompt for the Songs (*performed by*) dataset queries.

The mother tongue of Danielle Darrieux is

Figure 20: Example Prompt for the CounterFact (*mother tongue*) dataset queries.

Mahmoud Fawzi has a citizenship from

Figure 21: Example Prompt for the CounterFact (*citizenship*) dataset queries.

The headquarter of Monell Chemical Senses Center is located in

Figure 22: Example Prompt for the CounterFact (*headquarter location*) dataset queries.

User: Tell me a book that was written by Ernest Hemingway and that was published during 1923-1939
Assistant: The name of such a book is

Figure 23: Example Prompt for the Books (*author*,*published year*) dataset queries.

User: Is there a person who is a Nobel Prize Winner and who was born in the city of Cluny
Assistant: Yes, the person's name is

Figure 24: Example Prompt for the Nobel Winner (*won Nobel*,*born in city*) dataset queries.

User: Is there a word that starts with the letter u and ends with the letter d
Assistant: Yes, one such word is

Figure 25: Example Prompt for the Words (*starts with*,*ends with*) dataset queries.

| Popularity Interval | Sampled Teams |
| --- | --- |
| 0-10 | Zonguldak Kömürspor, Grêmio Esportivo Novorizontino |
| 10-20 | Sorrento Calcio 1945, FK Banat Zrenjanin |
| 20-30 | Valur, First Vienna FC |
| 30-40 | Kalmar FF, FK Budućnost Podgorica |
| 40-50 | Società Polisportiva Ars et Labor, Sheffield Wednesday F.C. |
| 50-60 | Granada CF, 1. FC Nürnberg |
| 60-70 | Nottingham Forest F.C., FK Džiugas Telšiai |
| 70-80 | Olympique de Marseille, FC Zenit Saint Petersburg |
| 80-90 | FC Porto, Club Social y Deportivo Colo Colo |
| 90-140 | Real Madrid CF, Manchester United F.C. |

Table 4: **Football Teams in Each Popularity Interval.** Popularity indicates the number of site links on the WikiData page of the team.

| Model | Data | Constraint | Model Success | $Risk_{Bottom\ 20\%}(\Uparrow)$ | | | | | | $Risk_{Top\ 20\%}(\Downarrow)$ | | | | | | AUROC$(\Uparrow)$ | | | | | |
|---|---|---|---|---|---|---|---|---|---|---|---|---|---|---|---|---|---|---|---|---|---|
| | | | | CONFIDENCE | SAT-PROBE(A) | SAT-PROBE(a) | CONSTANT | POPULARITY | COMBINED | CONFIDENCE | SAT-PROBE(A) | SAT-PROBE(a) | CONSTANT | POPULARITY | COMBINED | CONFIDENCE | SAT-PROBE(A) | SAT-PROBE(a) | CONSTANT | POPULARITY | COMBINED |
| 7B | Basketball Players | *born in the year* | 0.18 | 0.96 ± 0.00 | 0.95 ± 0.00 | 0.95 ± 0.00 | 0.82 ± 0.00 | 0.92 ± 0.00 | 0.96 ± 0.00 | 0.46 ± 0.00 | 0.47 ± 0.00 | 0.47 ± 0.00 | 0.82 ± 0.00 | 0.56 ± 0.00 | 0.46 ± 0.00 | 0.82 ± 0.00 | 0.81 ± 0.00 | 0.81 ± 0.00 | 0.50 ± 0.00 | 0.74 ± 0.00 | 0.82 ± 0.00 |
| 7B | CounterFact - Cit | *Citizenship* | 0.63 | 0.45 ± 0.01 | 0.66 ± 0.02 | 0.74 ± 0.01 | 0.38 ± 0.01 | 0.38 ± 0.01 | 0.71 ± 0.01 | 0.29 ± 0.01 | 0.19 ± 0.01 | 0.14 ± 0.01 | 0.38 ± 0.01 | 0.38 ± 0.01 | 0.15 ± 0.01 | 0.51 ± 0.01 | 0.71 ± 0.01 | 0.76 ± 0.01 | 0.50 ± 0.00 | 0.50 ± 0.00 | 0.76 ± 0.01 |
| 7B | CounterFact - HQ | *Headquarter Location* | 0.61 | 0.75 ± 0.01 | 0.81 ± 0.01 | 0.83 ± 0.01 | 0.38 ± 0.02 | 0.38 ± 0.02 | 0.87 ± 0.01 | 0.09 ± 0.01 | 0.20 ± 0.01 | 0.19 ± 0.00 | 0.38 ± 0.02 | 0.38 ± 0.02 | 0.08 ± 0.01 | 0.77 ± 0.01 | 0.75 ± 0.01 | 0.76 ± 0.01 | 0.50 ± 0.00 | 0.50 ± 0.00 | 0.84 ± 0.00 |
| 7B | CounterFact - MT | *Mother Tongue* | 0.48 | 0.98 ± 0.00 | 1.00 ± 0.00 | 1.00 ± 0.00 | 0.51 ± 0.02 | 0.51 ± 0.02 | 0.99 ± 0.00 | 0.08 ± 0.01 | 0.03 ± 0.00 | 0.03 ± 0.01 | 0.51 ± 0.02 | 0.51 ± 0.02 | 0.02 ± 0.00 | 0.89 ± 0.00 | 0.98 ± 0.00 | 0.98 ± 0.00 | 0.50 ± 0.00 | 0.50 ± 0.00 | 0.98 ± 0.00 |
| 7B | Football Teams | *founded in the year* | 0.13 | 0.97 ± 0.00 | 0.96 ± 0.00 | 0.97 ± 0.00 | 0.88 ± 0.00 | 0.94 ± 0.00 | 0.97 ± 0.00 | 0.60 ± 0.00 | 0.63 ± 0.00 | 0.61 ± 0.00 | 0.88 ± 0.00 | 0.67 ± 0.00 | 0.58 ± 0.00 | 0.81 ± 0.00 | 0.79 ± 0.00 | 0.81 ± 0.00 | 0.50 ± 0.00 | 0.74 ± 0.00 | 0.83 ± 0.00 |
| 7B | Movies | *directed by* | 0.58 | 0.97 ± 0.00 | 0.87 ± 0.00 | 0.88 ± 0.00 | 0.42 ± 0.00 | 0.62 ± 0.00 | 0.96 ± 0.00 | 0.04 ± 0.00 | 0.15 ± 0.00 | 0.14 ± 0.00 | 0.42 ± 0.00 | 0.19 ± 0.00 | 0.05 ± 0.00 | 0.90 ± 0.00 | 0.80 ± 0.00 | 0.81 ± 0.00 | 0.50 ± 0.00 | 0.69 ± 0.00 | 0.90 ± 0.00 |
| 7B | Songs | *performed by* | 0.22 | 1.00 ± 0.00 | 1.00 ± 0.00 | 1.00 ± 0.00 | 0.77 ± 0.01 | 0.93 ± 0.00 | 1.00 ± 0.00 | 0.27 ± 0.00 | 0.38 ± 0.00 | 0.38 ± 0.00 | 0.77 ± 0.01 | 0.53 ± 0.00 | 0.27 ± 0.01 | 0.93 ± 0.00 | 0.88 ± 0.00 | 0.88 ± 0.00 | 0.50 ± 0.00 | 0.74 ± 0.00 | 0.93 ± 0.00 |
| 13B | Basketball Players | *born in the year* | 0.22 | 0.95 ± 0.00 | 0.95 ± 0.00 | 0.95 ± 0.00 | 0.78 ± 0.00 | 0.91 ± 0.00 | 0.95 ± 0.00 | 0.28 ± 0.00 | 0.32 ± 0.00 | 0.32 ± 0.00 | 0.78 ± 0.00 | 0.43 ± 0.00 | 0.31 ± 0.00 | 0.86 ± 0.00 | 0.85 ± 0.00 | 0.85 ± 0.00 | 0.50 ± 0.00 | 0.77 ± 0.00 | 0.86 ± 0.00 |
| 13B | CounterFact - Cit | *Citizenship* | 0.60 | 0.57 ± 0.01 | 0.69 ± 0.01 | 0.74 ± 0.01 | 0.41 ± 0.01 | 0.41 ± 0.01 | 0.74 ± 0.01 | 0.23 ± 0.01 | 0.15 ± 0.01 | 0.13 ± 0.01 | 0.41 ± 0.01 | 0.41 ± 0.01 | 0.14 ± 0.01 | 0.58 ± 0.00 | 0.72 ± 0.01 | 0.75 ± 0.00 | 0.50 ± 0.00 | 0.50 ± 0.00 | 0.76 ± 0.01 |
| 13B | CounterFact - HQ | *Headquarter Location* | 0.64 | 0.67 ± 0.01 | 0.68 ± 0.01 | 0.70 ± 0.01 | 0.33 ± 0.02 | 0.33 ± 0.02 | 0.82 ± 0.01 | 0.08 ± 0.01 | 0.23 ± 0.01 | 0.19 ± 0.01 | 0.33 ± 0.02 | 0.33 ± 0.02 | 0.08 ± 0.01 | 0.77 ± 0.00 | 0.70 ± 0.01 | 0.72 ± 0.01 | 0.50 ± 0.00 | 0.50 ± 0.00 | 0.82 ± 0.01 |
| 13B | CounterFact - MT | *Mother Tongue* | 0.46 | 0.91 ± 0.01 | 0.99 ± 0.00 | 0.99 ± 0.00 | 0.53 ± 0.01 | 0.53 ± 0.01 | 0.98 ± 0.00 | 0.13 ± 0.01 | 0.08 ± 0.01 | 0.07 ± 0.01 | 0.53 ± 0.01 | 0.53 ± 0.01 | 0.06 ± 0.00 | 0.80 ± 0.00 | 0.96 ± 0.00 | 0.96 ± 0.00 | 0.50 ± 0.00 | 0.50 ± 0.00 | 0.96 ± 0.00 |
| 13B | Football Teams | *founded in the year* | 0.18 | 0.97 ± 0.00 | 0.96 ± 0.00 | 0.96 ± 0.00 | 0.81 ± 0.00 | 0.94 ± 0.00 | 0.97 ± 0.00 | 0.47 ± 0.00 | 0.48 ± 0.00 | 0.49 ± 0.00 | 0.81 ± 0.00 | 0.51 ± 0.01 | 0.44 ± 0.00 | 0.83 ± 0.00 | 0.81 ± 0.00 | 0.81 ± 0.00 | 0.50 ± 0.00 | 0.77 ± 0.00 | 0.88 ± 0.00 |
| 13B | Movies | *directed by* | 0.71 | 0.83 ± 0.00 | 0.70 ± 0.00 | 0.70 ± 0.00 | 0.29 ± 0.00 | 0.45 ± 0.00 | 0.83 ± 0.00 | 0.04 ± 0.00 | 0.09 ± 0.00 | 0.09 ± 0.00 | 0.29 ± 0.00 | 0.13 ± 0.00 | 0.04 ± 0.00 | 0.87 ± 0.00 | 0.79 ± 0.00 | 0.79 ± 0.00 | 0.50 ± 0.00 | 0.67 ± 0.00 | 0.88 ± 0.00 |
| 13B | Songs | *performed by* | 0.24 | 0.88 ± 0.00 | 0.73 ± 0.00 | 0.73 ± 0.00 | 0.76 ± 0.01 | 0.59 ± 0.00 | 0.88 ± 0.00 | 0.23 ± 0.01 | 0.35 ± 0.00 | 0.35 ± 0.01 | 0.76 ± 0.01 | 0.52 ± 0.00 | 0.25 ± 0.01 | 0.91 ± 0.00 | 0.88 ± 0.00 | 0.88 ± 0.00 | 0.50 ± 0.00 | 0.71 ± 0.00 | 0.92 ± 0.00 |
| 70B | Basketball Players | *born in the year* | 0.66 | 0.88 ± 0.00 | 0.73 ± 0.00 | 0.73 ± 0.00 | 0.34 ± 0.00 | 0.59 ± 0.00 | 0.88 ± 0.00 | 0.01 ± 0.00 | 0.06 ± 0.00 | 0.07 ± 0.00 | 0.34 ± 0.00 | 0.07 ± 0.00 | 0.01 ± 0.00 | 0.91 ± 0.00 | 0.81 ± 0.00 | 0.81 ± 0.00 | 0.50 ± 0.00 | 0.74 ± 0.00 | 0.91 ± 0.00 |
| 70B | CounterFact - Cit | *Citizenship* | 0.69 | 0.47 ± 0.01 | 0.69 ± 0.01 | 0.70 ± 0.01 | 0.32 ± 0.01 | 0.32 ± 0.01 | 0.58 ± 0.02 | 0.16 ± 0.01 | 0.10 ± 0.01 | 0.10 ± 0.01 | 0.32 ± 0.01 | 0.32 ± 0.01 | 0.16 ± 0.01 | 0.58 ± 0.00 | 0.78 ± 0.01 | 0.78 ± 0.01 | 0.50 ± 0.00 | 0.50 ± 0.00 | 0.70 ± 0.01 |
| 70B | CounterFact - HQ | *Headquarter Location* | 0.66 | 0.61 ± 0.01 | 0.69 ± 0.01 | 0.66 ± 0.01 | 0.31 ± 0.02 | 0.31 ± 0.02 | 0.69 ± 0.01 | 0.04 ± 0.01 | 0.13 ± 0.01 | 0.14 ± 0.02 | 0.31 ± 0.02 | 0.31 ± 0.02 | 0.06 ± 0.01 | 0.74 ± 0.00 | 0.74 ± 0.01 | 0.73 ± 0.01 | 0.50 ± 0.00 | 0.50 ± 0.00 | 0.77 ± 0.01 |
| 70B | CounterFact - MT | *Mother Tongue* | 0.46 | 0.94 ± 0.00 | 1.00 ± 0.00 | 1.00 ± 0.00 | 0.54 ± 0.01 | 0.54 ± 0.01 | 1.00 ± 0.00 | 0.13 ± 0.01 | 0.03 ± 0.00 | 0.02 ± 0.01 | 0.54 ± 0.01 | 0.54 ± 0.01 | 0.04 ± 0.01 | 0.82 ± 0.00 | 0.98 ± 0.00 | 0.97 ± 0.00 | 0.50 ± 0.00 | 0.50 ± 0.00 | 0.97 ± 0.00 |
| 70B | Football Teams | *founded in the year* | 0.38 | 0.95 ± 0.00 | 0.91 ± 0.00 | 0.90 ± 0.00 | 0.62 ± 0.01 | 0.86 ± 0.00 | 0.95 ± 0.00 | 0.24 ± 0.00 | 0.28 ± 0.00 | 0.28 ± 0.01 | 0.62 ± 0.01 | 0.27 ± 0.00 | 0.18 ± 0.00 | 0.83 ± 0.00 | 0.77 ± 0.00 | 0.77 ± 0.00 | 0.50 ± 0.00 | 0.76 ± 0.00 | 0.85 ± 0.00 |
| 70B | Movies | *directed by* | 0.86 | 0.34 ± 0.00 | 0.34 ± 0.00 | 0.34 ± 0.00 | 0.14 ± 0.00 | 0.17 ± 0.00 | 0.39 ± 0.00 | 0.04 ± 0.00 | 0.05 ± 0.00 | 0.03 ± 0.00 | 0.14 ± 0.00 | 0.10 ± 0.00 | 0.03 ± 0.00 | 0.75 ± 0.00 | 0.73 ± 0.00 | 0.75 ± 0.00 | 0.50 ± 0.00 | 0.55 ± 0.00 | 0.80 ± 0.00 |
| 70B | Songs | *performed by* | 0.44 | 0.97 ± 0.00 | 0.89 ± 0.01 | 0.89 ± 0.01 | 0.56 ± 0.01 | 0.71 ± 0.01 | 0.96 ± 0.00 | 0.17 ± 0.00 | 0.20 ± 0.01 | 0.18 ± 0.01 | 0.56 ± 0.01 | 0.35 ± 0.01 | 0.15 ± 0.01 | 0.85 ± 0.00 | 0.79 ± 0.00 | 0.80 ± 0.00 | 0.50 ± 0.00 | 0.65 ± 0.00 | 0.85 ± 0.00 |

Table 5: **Predicting factual errors for single-constraint queries** (⇑) indicates higher is better, and (⇓) indicates lower is better. We repeat the experiments with 10 random seeds where the randomness is over the train and test splits. We do not have the popularity numbers for the CounterFact dataset. Standard means ± standard errors across 10 random seeds are reported in each cell.

| Model | Data | Constraint | Model Success | $Risk_{Bottom\ 20\%}(\Uparrow)$ | | | | | $Risk_{Top\ 20\%}(\Downarrow)$ | | | | | AUROC$(\Uparrow)$ | | | | |
|---|---|---|---|---|---|---|---|---|---|---|---|---|---|---|---|---|---|---|
| | | | | SAT-PROBE(a) | SAT-PROBE(A) | CONFIDENCE | CONSTANT | COMBINED | SAT-PROBE(a) | SAT-PROBE(A) | CONFIDENCE | CONSTANT | COMBINED | SAT-PROBE(a) | SAT-PROBE(A) | CONFIDENCE | CONSTANT | COMBINED |
| 7B | Books | Overall | 0.13 | 0.99 ± 0.00 | 0.99 ± 0.00 | 1.00 ± 0.00 | 0.91 ± 0.01 | 1.00 ± 0.00 | 0.74 ± 0.01 | 0.77 ± 0.01 | 0.80 ± 0.01 | 0.91 ± 0.01 | 0.78 ± 0.01 | 0.78 ± 0.01 | 0.77 ± 0.01 | 0.73 ± 0.01 | 0.50 ± 0.00 | 0.82 ± 0.01 |
| 7B | Nobel Winner | Overall | 0.12 | 0.97 ± 0.00 | 0.98 ± 0.00 | 0.98 ± 0.00 | 0.95 ± 0.00 | 0.98 ± 0.00 | 0.91 ± 0.01 | 0.90 ± 0.01 | 0.88 ± 0.01 | 0.95 ± 0.01 | 0.85 ± 0.01 | 0.61 ± 0.01 | 0.66 ± 0.01 | 0.65 ± 0.02 | 0.50 ± 0.00 | 0.78 ± 0.01 |
| 7B | Words | Overall | 0.29 | 0.85 ± 0.01 | 0.90 ± 0.01 | 0.84 ± 0.01 | 0.79 ± 0.01 | 0.90 ± 0.01 | 0.71 ± 0.01 | 0.69 ± 0.01 | 0.70 ± 0.01 | 0.79 ± 0.01 | 0.65 ± 0.01 | 0.58 ± 0.01 | 0.63 ± 0.00 | 0.59 ± 0.01 | 0.50 ± 0.00 | 0.69 ± 0.01 |
| 13B | Books | Overall | 0.09 | 0.99 ± 0.00 | 1.00 ± 0.00 | 1.00 ± 0.00 | 0.91 ± 0.01 | 1.00 ± 0.00 | 0.83 ± 0.01 | 0.80 ± 0.01 | 0.80 ± 0.01 | 0.91 ± 0.01 | 0.80 ± 0.01 | 0.76 ± 0.00 | 0.82 ± 0.01 | 0.83 ± 0.00 | 0.50 ± 0.00 | 0.87 ± 0.00 |
| 13B | Nobel Winner | Overall | 0.13 | 0.98 ± 0.00 | 0.97 ± 0.00 | 0.99 ± 0.00 | 0.94 ± 0.01 | 0.98 ± 0.00 | 0.87 ± 0.00 | 0.86 ± 0.01 | 0.84 ± 0.01 | 0.94 ± 0.01 | 0.84 ± 0.00 | 0.66 ± 0.02 | 0.68 ± 0.01 | 0.74 ± 0.01 | 0.50 ± 0.00 | 0.74 ± 0.01 |
| 13B | Words | Overall | 0.32 | 0.86 ± 0.01 | 0.87 ± 0.01 | 0.82 ± 0.01 | 0.80 ± 0.02 | 0.87 ± 0.01 | 0.61 ± 0.01 | 0.61 ± 0.01 | 0.69 ± 0.01 | 0.80 ± 0.02 | 0.69 ± 0.01 | 0.63 ± 0.01 | 0.63 ± 0.01 | 0.57 ± 0.01 | 0.50 ± 0.00 | 0.69 ± 0.01 |
| 70B | Books | Overall | 0.17 | 0.98 ± 0.00 | 0.99 ± 0.00 | 0.97 ± 0.00 | 0.87 ± 0.00 | 0.98 ± 0.01 | 0.58 ± 0.01 | 0.55 ± 0.01 | 0.65 ± 0.01 | 0.87 ± 0.01 | 0.58 ± 0.01 | 0.81 ± 0.01 | 0.84 ± 0.01 | 0.76 ± 0.01 | 0.50 ± 0.00 | 0.86 ± 0.01 |
| 70B | Nobel Winner | Overall | 0.21 | 0.93 ± 0.01 | 0.94 ± 0.01 | 0.94 ± 0.01 | 0.84 ± 0.01 | 0.93 ± 0.01 | 0.67 ± 0.01 | 0.67 ± 0.01 | 0.63 ± 0.01 | 0.84 ± 0.01 | 0.69 ± 0.01 | 0.70 ± 0.01 | 0.71 ± 0.01 | 0.75 ± 0.01 | 0.50 ± 0.00 | 0.73 ± 0.01 |
| 70B | Words | Overall | 0.36 | 0.91 ± 0.01 | 0.91 ± 0.01 | 0.87 ± 0.01 | 0.85 ± 0.01 | 0.90 ± 0.01 | 0.38 ± 0.01 | 0.37 ± 0.01 | 0.53 ± 0.01 | 0.85 ± 0.01 | 0.40 ± 0.01 | 0.76 ± 0.01 | 0.77 ± 0.00 | 0.66 ± 0.01 | 0.50 ± 0.00 | 0.80 ± 0.01 |

Table 6: **Predicting factual errors for multi-constraint queries** (⇑) indicates higher is better, and (⇓) indicates lower is better. We repeat the experiments with 10 random seeds where the randomness is over the train and test splits. Standard means ± standard errors across 10 random seeds are reported in each cell.

| Model | Data | Constraint | Model Success | Risk_Bottom 20%(⇑) | | | Risk_Top 20%(⇓) | | | AUROC(⇑) | | |
|---|---|---|---|---|---|---|---|---|---|---|---|---|
| | | | | SAT-PROBE(a) | SAT-PROBE(A) | CONSTANT | SAT-PROBE(a) | SAT-PROBE(A) | CONSTANT | SAT-PROBE(a) | SAT-PROBE(A) | CONSTANT |
| 7B | Books | *author* | 0.38 | 0.95 ± 0.01 | 0.97 ± 0.01 | 0.61 ± 0.01 | 0.16 ± 0.01 | 0.25 ± 0.01 | 0.61 ± 0.01 | 0.85 ± 0.00 | 0.84 ± 0.00 | 0.50 ± 0.00 |
| 7B | Books | *published year* | 0.11 | 0.97 ± 0.00 | 0.98 ± 0.00 | 0.90 ± 0.01 | 0.81 ± 0.01 | 0.79 ± 0.01 | 0.90 ± 0.01 | 0.68 ± 0.01 | 0.71 ± 0.01 | 0.50 ± 0.00 |
| 7B | Nobel Winner | *born in city* | 0.06 | 0.95 ± 0.01 | 0.97 ± 0.01 | 0.94 ± 0.01 | 0.91 ± 0.01 | 0.86 ± 0.01 | 0.94 ± 0.01 | 0.57 ± 0.01 | 0.68 ± 0.01 | 0.50 ± 0.00 |
| 7B | Nobel Winner | *won Nobel* | 0.62 | 0.58 ± 0.01 | 0.66 ± 0.01 | 0.42 ± 0.01 | 0.21 ± 0.02 | 0.14 ± 0.01 | 0.42 ± 0.01 | 0.65 ± 0.01 | 0.72 ± 0.01 | 0.50 ± 0.00 |
| 7B | Words | *ends with* | 0.23 | 0.84 ± 0.01 | 0.90 ± 0.01 | 0.77 ± 0.01 | 0.70 ± 0.01 | 0.69 ± 0.01 | 0.77 ± 0.01 | 0.58 ± 0.01 | 0.62 ± 0.01 | 0.50 ± 0.00 |
| 7B | Words | *starts with* | 0.93 | 0.22 ± 0.01 | 0.27 ± 0.01 | 0.12 ± 0.01 | 0.01 ± 0.00 | 0.01 ± 0.00 | 0.12 ± 0.01 | 0.81 ± 0.01 | 0.85 ± 0.01 | 0.50 ± 0.00 |
| 13B | Books | *author* | 0.25 | 0.96 ± 0.00 | 0.98 ± 0.01 | 0.71 ± 0.01 | 0.49 ± 0.01 | 0.33 ± 0.02 | 0.71 ± 0.01 | 0.77 ± 0.00 | 0.87 ± 0.01 | 0.50 ± 0.00 |
| 13B | Books | *published year* | 0.08 | 0.98 ± 0.00 | 0.99 ± 0.00 | 0.91 ± 0.01 | 0.85 ± 0.01 | 0.86 ± 0.01 | 0.91 ± 0.01 | 0.68 ± 0.01 | 0.70 ± 0.01 | 0.50 ± 0.00 |
| 13B | Nobel Winner | *born in city* | 0.08 | 0.96 ± 0.02 | 0.96 ± 0.01 | 0.94 ± 0.01 | 0.86 ± 0.01 | 0.86 ± 0.01 | 0.94 ± 0.01 | 0.67 ± 0.03 | 0.66 ± 0.02 | 0.50 ± 0.00 |
| 13B | Nobel Winner | *won Nobel* | 0.64 | 0.73 ± 0.01 | 0.72 ± 0.01 | 0.44 ± 0.01 | 0.14 ± 0.01 | 0.11 ± 0.01 | 0.44 ± 0.01 | 0.75 ± 0.01 | 0.77 ± 0.00 | 0.50 ± 0.00 |
| 13B | Words | *ends with* | 0.27 | 0.83 ± 0.01 | 0.83 ± 0.01 | 0.79 ± 0.02 | 0.63 ± 0.01 | 0.61 ± 0.01 | 0.79 ± 0.02 | 0.61 ± 0.01 | 0.61 ± 0.01 | 0.50 ± 0.00 |
| 13B | Words | *starts with* | 0.89 | 0.45 ± 0.01 | 0.44 ± 0.02 | 0.19 ± 0.01 | 0.01 ± 0.00 | 0.01 ± 0.00 | 0.19 ± 0.01 | 0.92 ± 0.00 | 0.93 ± 0.01 | 0.50 ± 0.00 |
| 70B | Books | *author* | 0.32 | 0.95 ± 0.01 | 0.96 ± 0.01 | 0.73 ± 0.01 | 0.25 ± 0.01 | 0.24 ± 0.01 | 0.73 ± 0.01 | 0.82 ± 0.01 | 0.85 ± 0.01 | 0.50 ± 0.00 |
| 70B | Books | *published year* | 0.16 | 0.98 ± 0.00 | 0.98 ± 0.00 | 0.87 ± 0.01 | 0.61 ± 0.01 | 0.56 ± 0.01 | 0.87 ± 0.01 | 0.78 ± 0.01 | 0.82 ± 0.01 | 0.50 ± 0.00 |
| 70B | Nobel Winner | *born in city* | 0.16 | 0.94 ± 0.00 | 0.95 ± 0.01 | 0.83 ± 0.01 | 0.66 ± 0.01 | 0.66 ± 0.01 | 0.83 ± 0.01 | 0.71 ± 0.01 | 0.72 ± 0.01 | 0.50 ± 0.00 |
| 70B | Nobel Winner | *won Nobel* | 0.74 | 0.38 ± 0.01 | 0.38 ± 0.02 | 0.28 ± 0.01 | 0.18 ± 0.01 | 0.18 ± 0.01 | 0.28 ± 0.01 | 0.61 ± 0.01 | 0.60 ± 0.01 | 0.50 ± 0.00 |
| 70B | Words | *ends with* | 0.32 | 0.88 ± 0.01 | 0.90 ± 0.01 | 0.82 ± 0.01 | 0.38 ± 0.01 | 0.37 ± 0.01 | 0.82 ± 0.01 | 0.73 ± 0.01 | 0.75 ± 0.01 | 0.50 ± 0.00 |
| 70B | Words | *starts with* | 0.85 | 0.45 ± 0.01 | 0.47 ± 0.01 | 0.28 ± 0.01 | 0.01 ± 0.00 | 0.00 ± 0.00 | 0.28 ± 0.01 | 0.87 ± 0.00 | 0.88 ± 0.00 | 0.50 ± 0.00 |

Table 7: **Predicting the failure to satisfy individual constraints in multi-constraint settings.** (⇑) indicates higher is better, and (⇓) indicates lower is better. Here, we train individual failure predictors for each constraint and give the failure prediction metrics across all models, datasets, and constraints. Standard means ± standard errors across 10 random seeds are reported in each cell.

| Model | Data | Constraint | Model Success | Risk_Bottom 20%(⇑) | | | | | Risk_Top 20%(⇓) | | | | | AUROC(⇑) | | | | |
|---|---|---|---|---|---|---|---|---|---|---|---|---|---|---|---|---|---|---|
| | | | | CONFIDENCE | SAT-PROBE(A) | SAT-PROBE(a) | CONSTANT | POPULARITY | CONFIDENCE | SAT-PROBE(A) | SAT-PROBE(a) | CONSTANT | POPULARITY | CONFIDENCE | SAT-PROBE(A) | SAT-PROBE(a) | CONSTANT | POPULARITY |
| 7B | Basketball Players | *born in the year* | 0.18 | 0.96 ± 0.00 | 0.95 ± 0.00 | 0.94 ± 0.00 | 0.83 ± 0.00 | 0.92 ± 0.00 | 0.46 ± 0.01 | 0.47 ± 0.01 | 0.47 ± 0.00 | 0.83 ± 0.00 | 0.56 ± 0.00 | 0.82 ± 0.00 | 0.80 ± 0.00 | 0.80 ± 0.00 | 0.50 ± 0.00 | 0.74 ± 0.00 |
| 7B | CounterFact | *Citizenship* | 0.63 | 0.45 ± 0.01 | 0.65 ± 0.01 | 0.61 ± 0.02 | 0.38 ± 0.02 | - | 0.29 ± 0.02 | 0.24 ± 0.02 | 0.23 ± 0.03 | 0.38 ± 0.02 | - | 0.51 ± 0.01 | 0.68 ± 0.01 | 0.67 ± 0.01 | 0.50 ± 0.00 | - |
| 7B | CounterFact | *Headquarter Location* | 0.61 | 0.75 ± 0.01 | 0.72 ± 0.01 | 0.71 ± 0.02 | 0.39 ± 0.03 | - | 0.11 ± 0.01 | 0.22 ± 0.01 | 0.23 ± 0.02 | 0.39 ± 0.03 | - | 0.76 ± 0.00 | 0.72 ± 0.01 | 0.71 ± 0.01 | 0.50 ± 0.00 | - |
| 7B | CounterFact | *Mother Tongue* | 0.48 | 0.98 ± 0.00 | 1.00 ± 0.00 | 0.99 ± 0.00 | 0.51 ± 0.03 | - | 0.08 ± 0.01 | 0.03 ± 0.01 | 0.02 ± 0.00 | 0.51 ± 0.03 | - | 0.89 ± 0.01 | 0.98 ± 0.00 | 0.97 ± 0.00 | 0.50 ± 0.00 | - |
| 7B | Football Teams | *founded in the year* | 0.13 | 0.97 ± 0.00 | 0.97 ± 0.00 | 0.97 ± 0.00 | 0.87 ± 0.00 | 0.94 ± 0.00 | 0.60 ± 0.00 | 0.62 ± 0.00 | 0.63 ± 0.00 | 0.87 ± 0.00 | 0.66 ± 0.01 | 0.81 ± 0.00 | 0.79 ± 0.00 | 0.80 ± 0.00 | 0.50 ± 0.00 | 0.74 ± 0.00 |
| 7B | Movies | *directed by* | 0.58 | 0.97 ± 0.00 | 0.86 ± 0.00 | 0.85 ± 0.00 | 0.42 ± 0.01 | 0.62 ± 0.00 | 0.04 ± 0.00 | 0.16 ± 0.01 | 0.16 ± 0.01 | 0.42 ± 0.01 | 0.19 ± 0.00 | 0.90 ± 0.00 | 0.79 ± 0.00 | 0.79 ± 0.00 | 0.50 ± 0.00 | 0.69 ± 0.00 |
| 7B | Songs | *performed by* | 0.22 | 1.00 ± 0.00 | 0.99 ± 0.00 | 0.99 ± 0.00 | 0.77 ± 0.01 | 0.93 ± 0.01 | 0.28 ± 0.00 | 0.39 ± 0.01 | 0.40 ± 0.01 | 0.77 ± 0.01 | 0.53 ± 0.00 | 0.93 ± 0.00 | 0.86 ± 0.00 | 0.86 ± 0.00 | 0.50 ± 0.00 | 0.74 ± 0.00 |
| 13B | Basketball Players | *born in the year* | 0.22 | 0.95 ± 0.00 | 0.93 ± 0.00 | 0.93 ± 0.00 | 0.78 ± 0.00 | 0.91 ± 0.00 | 0.29 ± 0.00 | 0.33 ± 0.00 | 0.34 ± 0.00 | 0.78 ± 0.00 | 0.43 ± 0.01 | 0.86 ± 0.00 | 0.84 ± 0.00 | 0.83 ± 0.00 | 0.50 ± 0.00 | 0.76 ± 0.00 |
| 13B | CounterFact | *Citizenship* | 0.60 | 0.58 ± 0.01 | 0.60 ± 0.02 | 0.61 ± 0.01 | 0.43 ± 0.02 | - | 0.22 ± 0.01 | 0.24 ± 0.02 | 0.25 ± 0.02 | 0.43 ± 0.02 | - | 0.58 ± 0.01 | 0.66 ± 0.02 | 0.66 ± 0.01 | 0.50 ± 0.00 | - |
| 13B | CounterFact | *Headquarter Location* | 0.64 | 0.68 ± 0.01 | 0.56 ± 0.01 | 0.53 ± 0.02 | 0.32 ± 0.04 | - | 0.09 ± 0.01 | 0.19 ± 0.01 | 0.21 ± 0.02 | 0.32 ± 0.04 | - | 0.77 ± 0.01 | 0.66 ± 0.01 | 0.64 ± 0.01 | 0.50 ± 0.00 | - |
| 13B | CounterFact | *Mother Tongue* | 0.46 | 0.90 ± 0.01 | 0.99 ± 0.00 | 0.97 ± 0.00 | 0.53 ± 0.02 | - | 0.14 ± 0.01 | 0.06 ± 0.01 | 0.07 ± 0.01 | 0.53 ± 0.02 | - | 0.80 ± 0.00 | 0.95 ± 0.00 | 0.95 ± 0.00 | 0.50 ± 0.00 | - |
| 13B | Football Teams | *founded in the year* | 0.18 | 0.97 ± 0.00 | 0.95 ± 0.00 | 0.95 ± 0.00 | 0.82 ± 0.00 | 0.94 ± 0.00 | 0.47 ± 0.00 | 0.52 ± 0.00 | 0.52 ± 0.00 | 0.82 ± 0.00 | 0.52 ± 0.01 | 0.83 ± 0.00 | 0.79 ± 0.00 | 0.78 ± 0.00 | 0.50 ± 0.00 | 0.77 ± 0.01 |
| 13B | Movies | *directed by* | 0.71 | 0.83 ± 0.00 | 0.66 ± 0.00 | 0.66 ± 0.00 | 0.30 ± 0.00 | 0.45 ± 0.00 | 0.04 ± 0.00 | 0.10 ± 0.00 | 0.11 ± 0.00 | 0.30 ± 0.00 | 0.12 ± 0.00 | 0.87 ± 0.00 | 0.76 ± 0.00 | 0.76 ± 0.00 | 0.50 ± 0.00 | 0.67 ± 0.00 |
| 13B | Songs | *performed by* | 0.24 | 1.00 ± 0.00 | 0.97 ± 0.00 | 0.98 ± 0.00 | 0.75 ± 0.01 | 0.89 ± 0.01 | 0.24 ± 0.00 | 0.40 ± 0.01 | 0.39 ± 0.01 | 0.75 ± 0.01 | 0.52 ± 0.00 | 0.93 ± 0.00 | 0.83 ± 0.00 | 0.84 ± 0.00 | 0.50 ± 0.00 | 0.71 ± 0.00 |
| 70B | Basketball Players | *born in the year* | 0.66 | 0.87 ± 0.00 | 0.62 ± 0.00 | 0.58 ± 0.00 | 0.34 ± 0.00 | 0.58 ± 0.00 | 0.01 ± 0.00 | 0.09 ± 0.00 | 0.09 ± 0.00 | 0.34 ± 0.00 | 0.07 ± 0.00 | 0.91 ± 0.00 | 0.76 ± 0.00 | 0.73 ± 0.00 | 0.50 ± 0.00 | 0.74 ± 0.00 |
| 70B | CounterFact | *Citizenship* | 0.69 | 0.48 ± 0.01 | 0.60 ± 0.02 | 0.56 ± 0.02 | 0.32 ± 0.02 | - | 0.16 ± 0.01 | 0.15 ± 0.01 | 0.16 ± 0.01 | 0.32 ± 0.02 | - | 0.58 ± 0.01 | 0.71 ± 0.01 | 0.70 ± 0.01 | 0.50 ± 0.00 | - |
| 70B | CounterFact | *Headquarter Location* | 0.66 | 0.61 ± 0.02 | 0.55 ± 0.01 | 0.55 ± 0.02 | 0.30 ± 0.04 | - | 0.04 ± 0.01 | 0.18 ± 0.02 | 0.23 ± 0.02 | 0.30 ± 0.04 | - | 0.74 ± 0.00 | 0.66 ± 0.01 | 0.65 ± 0.02 | 0.50 ± 0.00 | - |
| 70B | CounterFact | *Mother Tongue* | 0.46 | 0.95 ± 0.01 | 1.00 ± 0.00 | 1.00 ± 0.00 | 0.53 ± 0.03 | - | 0.13 ± 0.01 | 0.06 ± 0.02 | 0.08 ± 0.01 | 0.53 ± 0.03 | - | 0.82 ± 0.00 | 0.96 ± 0.00 | 0.95 ± 0.00 | 0.50 ± 0.00 | - |
| 70B | Football Teams | *founded in the year* | 0.38 | 0.95 ± 0.00 | 0.84 ± 0.00 | 0.80 ± 0.00 | 0.62 ± 0.00 | 0.86 ± 0.00 | 0.04 ± 0.00 | 0.37 ± 0.00 | 0.39 ± 0.01 | 0.62 ± 0.00 | 0.28 ± 0.00 | 0.83 ± 0.00 | 0.71 ± 0.00 | 0.68 ± 0.00 | 0.50 ± 0.00 | 0.76 ± 0.00 |
| 70B | Movies | *directed by* | 0.86 | 0.34 ± 0.01 | 0.25 ± 0.01 | 0.23 ± 0.00 | 0.14 ± 0.01 | 0.17 ± 0.00 | 0.04 ± 0.00 | 0.06 ± 0.00 | 0.06 ± 0.00 | 0.14 ± 0.01 | 0.10 ± 0.01 | 0.75 ± 0.00 | 0.67 ± 0.00 | 0.66 ± 0.01 | 0.50 ± 0.00 | 0.56 ± 0.01 |
| 70B | Songs | *performed by* | 0.44 | 0.97 ± 0.00 | 0.76 ± 0.01 | 0.73 ± 0.01 | 0.55 ± 0.01 | 0.71 ± 0.01 | 0.16 ± 0.01 | 0.27 ± 0.01 | 0.29 ± 0.01 | 0.55 ± 0.01 | 0.35 ± 0.01 | 0.85 ± 0.00 | 0.69 ± 0.00 | 0.67 ± 0.00 | 0.50 ± 0.00 | 0.65 ± 0.01 |

Table 8: **Using a single predictor for all constraints in single-constraint queries.** (⇑) indicates higher is better, and (⇓) indicates lower is better. We repeat the experiments with 10 random seeds where the randomness is over the train and test splits. We present the standard errors next to the average results across all experiments. We do not have the popularity numbers for the CounterFact dataset. Standard means ± standard errors across 5 random seeds are reported in each cell.