# OpenReview forum: "Attention Satisfies: A Constraint-Satisfaction Lens on Factual Errors of Language Models"
_ICLR.cc/2024/Conference — ICLR 2024 poster_

### Official Review · Reviewer_MD6P · 2023-11-01

**Soundness:** 3 good
**Presentation:** 4 excellent
**Contribution:** 3 good
**Rating:** 6
**Confidence:** 4

**Summary:**

This paper presents an interesting finding that the magnitude of attention contribution to “constraint entities” (something like key entities associated with the prediction) is well-correlated with popularity of the constraint entity, the constrainedness of the query, and more importantly, the correctness of the LLM prediction. The authors define metrics to quantify such attention contribution and expect to use the metric to predict factual errors of LLM predictions. On experiments across multiple datasets, the proposed method demonstrates comparable factual error prediction performance to the confidence (i.e. logits) baseline.

**Strengths:**

1. This paper presents interesting and novel findings of a phenomenon in the hidden representations of LLMs, that the magnitude of attention contribution is correlated with factualness of the prediction. It may inspire future researchers to discover and probe more interpretable inner patterns of LLMs, which could potentially help researchers understand and explain models’ behaviours.
2. The authors perform comprehensive analysis to verify the correlation of attention contribution with various properties such as popularity, constrainedness, and factualness. These analyses are convincing and the different types of visualization are informative and efficient to convey the messages.
3. The paper is very well-written.

**Weaknesses:**

1. Although the findings are interesting to know, I doubt the practicality of the proposed method and the contribution it makes from a practical perspective:
    a. Attention contribution is computed assuming access to the constraint entities, which are not available in most of cases
    b. The proposed method is not better on factual error prediction than the simple confidence baseline. It seems to me it does not bring any new information beyond the confidence score

2. I think this paper focuses on and only studies a narrow notion of factuality, where constraint entities exist as in a world knowledge domain. It is unclear whether attention contribution is a valid metric to explain other factual errors, such as mathematical errors and inference errors where the traditional notion of “entities” may not always exist. This is important since researchers talk about factuality and hallucination actually in a quite broad scope in the LLM context nowadays, not limited to entity-based knowledge.

3. From Figure 14-23, all the datasets look simple and toy. Although it appears that the experiments are conducted on a diverse range of datasets, these datasets turn out to be very similar style-wise – they are very similar QA tasks with one-entity answers. While I understand that it is difficult to perform experiments on more realistic data since the constraint entities will become unavailable, I am not sure how general the proposed findings are for LLMs given these toy datasets.

**Questions:**

1. In section 5.1 the authors mention fitting Logistic Regression, does that mean you need a training dataset with labels to learn the parameters w and b? If so, how large is the training data?

---

> ### Author Response · Authors · 2023-11-15
> **Response to Reviewer MD6P 1/2**
>
> Dear Reviewer MD6P,
>
> Thank you so much for your detailed insights and the time you took to review our paper. We are extremely excited that you find our findings novel, and find that our work can inspire many future works! At a high level, we provide new results that
> - (i) we show that the errors that attention and confidence make are different
> - (ii) this lets us come up with a method that combines confidence and attention, which works better than confidence alone.
>
> We further incorporated the suggestions, clarified the contribution of our framework compared to the existing literature, and resolved ambiguities in the text.
>
> 1. *The proposed method is not better on factual error prediction than the simple confidence baseline. It seems to me it does not bring any new information beyond the confidence score*
>
>  **Do confidence and attention result in the same predictions? Combining Attention and Confidence**: In Appendix E Figure 15, we compare the predictions by Attention vs Confidence for all three models in the Llama family. Overall, while they do correlate, there are many cases where both predictors disagree. For instance, the model can be overconfident and attention can correctly identify factual errors (see top left). Because the errors are not identical, we can combine them to build a better system.
>
> We extended our results with a *Combined* predictor, where we add confidence in addition to the attention features, and perform logistic regression. This model mostly performs the best overall, in terms of AUROC (Figure 6), and risk in either tail (bottom 20% and top 20% most confident predictions, Figures 9 and 10).
>
>  **Why is the finding that model attention can be used to predict constraint satisfaction and is comparable to confidence important\interesting?** We would like to highlight that the LLM is optimized using confidence as the objective (e.g. probability of the next token). We find it to be an extremely interesting finding that the process that the LLM uses to produce an output, without information about the objects it is acting on (predicted probability distribution), gives almost as much information as the output. We believe our findings could lead to several follow-up works in studying factual errors and the model’s reliability. Why do we have more attention to popular entities? How can we manipulate attention to increase constraint satisfaction and steer the model behavior? Our framework and findings open interesting questions and can pave the way for future research in this area.
>
> **Important difference between attention and confidence:** As we emphasize in the paper, note that confidence is only associated with the generated tokens and cannot be traced or mapped back to individual constraints for queries with multiple constraints. This is a crucial functional advantage of SAT-Probe and becomes even more important as the query complexity increases.
>
> 2. *Although the findings are interesting to know, I doubt the practicality of the proposed method*
>
> **Constraint availability:** We agree that our experiments focus on settings that assume access to constraints. Nevertheless, constraint extraction from user queries is an interesting language task to study by itself and can be considered as a pre-processing step prior to any factual verification method. Relevant methods may rely on entity recognition, previous work [a,b] on parsing declarative queries from natural text, or innovations in chain-of-thought methods that first extract main constraints from queries before generating factual answers. We added this note in our conclusion section.
>
> [a] Elgohary, Ahmed, Saghar Hosseini, and Ahmed Hassan Awadallah. "Speak to your parser: Interactive text-to-SQL with natural language feedback." arXiv preprint arXiv:2005.02539 (2020).
>
> [b] Yaghmazadeh, N., Wang, Y., Dillig, I., & Dillig, T. (2017). SQLizer: query synthesis from natural language. Proceedings of the ACM on Programming Languages, 1(OOPSLA), 1-26.

---

> ### Author Response · Authors · 2023-11-15
> **Response to Reviewer MD6P 2/2**
>
> 3. *I think this paper focuses on and only studies a narrow notion of factuality, where constraint entities exist as in a world knowledge domain*
>
> **Generality of the framework:** We agree with the reviewer that the CSP framework does not cover all potential queries to LLMs. Our work lays out a framework for mechanistic interpretability to analyze a broader class of queries, thus contributing to the literature: **existing work** in factuality and mechanistic interpretability in LLMs (e.g. Meng et al. NeurIPS 2022, Geva et al. EMNLP 2023) only considered single-constraint queries with well-defined, yet restrictive (subject, relation, object) structure. While there are still limitations, it creates further opportunities regarding what we can understand with a mechanistic lens, in particular by directly mapping attention patterns of individual constraints to model performance. We would appreciate it if the reviewer could consider the context of the contribution here.
>
> Further, we do not only look at CounterFact and we expand the set of datasets further to questions that are verifiable and that can also be reused in future research. We further agree with the reviewer that there is more work to do around mathematical knowledge and reasoning, and we believe these are interesting avenues for future work. We added this to the conclusion section, thank you for your suggestions.
>
> 4. *From Figure 14-23, all the datasets look simple and toy.*
>
> As further shown in Table 1 and Figure 6a, our datasets also contain queries with double constraints (which were previously not covered in related work). Most of the related work on mechanistic interpretability for factual queries has used CounterFact (which we also study) for single-constraint queries. Here, we expanded the set of datasets, and hope that this can serve as a resource to the community to study more complex query forms.
>
> 5. *does that mean you need a training dataset with labels to learn the parameters w and b?*
>
> In each of the datasets, we split the dataset into 50% train -50% test splits and train on the training set. Thus, the number of data points used to train a probe is half of the number of queries in Table 1 when testing on respective datasets.
>
> **Overall**, we do believe we are studying the problem of factual error prediction from a novel point of view, our findings around attention and factual error prediction, datasets, and the framework could prove useful to the literature. If you also find similar value in this research, the future work that can build on our insights,  and contextualization in the literature, we would appreciate it if you could consider increasing your score. Thank you very much once again, and we are happy to follow up with any additional questions you may have!

---

> > ### Author Response · Authors · 2023-11-19
> > **We would love to hear from you!**
> >
> > Dear Reviewer MD6P,
> >
> > Once again thank you very much for your detailed review and the time you spent. As we are nearing the end of the discussion period, we would like to ask if the questions you raised have been addressed. We hope you find our responses useful and would love to engage with you further if there are any remaining points.
> >
> > We understand that the discussion period is short, and we sincerely appreciate your time and help!

---

> > > ### Comment · Reviewer_MD6P · 2023-11-22
> > > **Thanks for the response**
> > >
> > > Thank you for the response, and I appreciate that the authors add the results of a combined baseline.
> > >
> > > To clarify, I do acknowledge the strengths of this work and the contribution of probing the attention patterns for factuality, which was the main reason I gave a positive score of 6.
> > >
> > > However, my concerns on practicality remain -- the combined results mitigate this concern a bit, but the reliance on constraint availability and the toy setting/scope of factuality and evaluation datasets are limitations of this work. I understand that constraint extraction as a preprocessing step may be done well, but this argument is pretty weak without experimental evidence -- the experiments of this paper assume existence of "ground-truth" constraints, which is a rigorous condition that the confidence baseline does not require. Furthermore, I know that many factuality works are performing experiments on toy settings or datasets, especially before LLMs appear. However, as LLMs come out and release so much power, I really look forward to moves of this direction to more realistic settings, at least attempting to do so, for example in [1] (this work is concurrent and not particularly relevant, but they tried to evaluate on more realistic data).
> > >
> > > Again, I lean towards acceptance of this paper, yet it is just not exciting enough for me to further increase the score due to these concerns.
> > >
> > >
> > > [1] Chern et al. FacTool: Factuality Detection in Generative AI--A Tool Augmented Framework for Multi-Task and Multi-Domain Scenarios. 2023

---

### Official Review · Reviewer_m32R · 2023-11-03

**Soundness:** 3 good
**Presentation:** 3 good
**Contribution:** 2 fair
**Rating:** 6
**Confidence:** 3

**Summary:**

This work focuses on investigating characteristics of attention patterns when LLM makes factual errors or hallucinates. They show that entity tokens (defined as constraining tokens) in the query input exhibit strong correlation with the answer entity tokens. The authors had three key observations 1.) the more popular an entity-relation pair is (i.e., appears more frequently in corpus), LLM is less likely to be factually incorrect about it 2.) attention scores can serve as a way to measure LLM confidence and correctness and 3.) the larger the LLM the more the attention scores and attention scores correlate in a similar way across different size models.

**Strengths:**

1.) This work looks at mechanistic interpretability especially with respect to factor errors made by LLMs, which has not been studied before as per my knowledge.

2.) They do extensive analysis and show the effectiveness of their probing technique across different datasets and models.

3.) SAT-probe can also help predict failures by analyzing attention scores mid-training, which can be useful of debugging purposes.

**Weaknesses:**

None I can think of

**Questions:**

1.) The observation that "when the LLM is accurate, there is more attention to constraint token" suggests that when LLM is confident it is often right. How does this generalize in cases shown in prior work when LLMs confidently hallucinate? It would seem that attention scores would be high even when hallucinating. In your experience, is there any way to differentiate between attention scores (or some other variable) for correct vs incorrect confident answers.

2.) In Figure 5, it looks like attention is summed across layers, its unclear if the final score was normalized by number of layers? Is it possible that attention scores are more comparable between 7B and 13B as compared to 7B and 70B because of the number of layers scaling?

3.) Were there any insights as to how attention patterns vary across different layers, especially for different sized models? for e.g., higher attention scores for hallucinating tokens towards the later layers perhaps?

---

> ### Author Response · Authors · 2023-11-15
> **Response to Reviewer m32R**
>
> Dear Reviewer m32R,
>
> Thank you so much for your kind words and the time you took to review our paper. We are truly excited by your support, and we appreciate that you found no major weaknesses; and thanks for your insightful questions!
>
> At a high level, we provide new results that
> - (i) show that the errors that attention and confidence make are different
> - (ii) this lets us come up with a method that combines confidence and attention, which works better than confidence and attention alone.
>
> Below we respond to the individual points you raised.
>
> 1. *...suggests that when LLM is confident it is often right. How does this generalize in cases shown in prior work when LLMs confidently hallucinate?*, *In your experience, is there any way to differentiate between attention scores (or some other variable) for correct vs incorrect confident answers*
>
> **Differences between confidence vs attention scores**: In Appendix E Figure 15, we compare the predictions by Attention vs Confidence for all three models in the Llama family. Overall, while they do correlate, there are many cases where both predictors disagree. For instance, the model can be overconfident and attention can correctly identify factual errors (see top left). Thus, there is value in considering both predictors.
>
> **Combining attention and confidence:** We extended our results with a *Combined* predictor, where we add confidence in addition to the attention features, and perform logistic regression. This model mostly performs the best overall, in terms of AUROC (Figure 6), and risk in either tail (bottom 20% and top 20% most confident predictions, Figures 9 and 10).
>
> 2. *In Figure 5, it looks like attention is summed across layers, its unclear if the final score was normalized by number of layers?*
>
> **Normalization in Figure 5**: We sum the values across layers, and normalize all numbers by the maximum observed value as indicated in the caption of Figure 5. Indeed, for models with a larger number of layers, the unnormalized values are larger. We wanted the visualization to be more interpretable within the context of the model being investigated. We added further clarification to the caption (see in red), and thank you for raising this point.
>
> 3. *Were there any insights as to how attention patterns vary across different layers, especially for different-sized models? for e.g., higher attention scores for hallucinating tokens towards the later layers perhaps?*
>
> **Attention Across Layers**: Very interesting question! We find that scaling matters here. In Appendix Figure 7, we discuss the impact of early stopping. Interestingly, we observe that we can make failure predictions in the earlier layers as well as the later layers for smaller models (left two plots). However, for larger models, we need to observe the attention in the later layers to make better predictions. We believe one can hypothesize that attention scores in later layers are more predictive for the larger models than smaller models, based on these findings.  Other than this, we did not observe stronger findings around larger attention values in later layers indicating factual errors, but we believe our tools (e.g. tracking the dynamics of attention to constraints) can lead to future research on this question.
>
> **Overall**, we do believe we are studying the problem of factual error prediction from a novel point of view, our findings around attention and factual error prediction, datasets, and the framework could prove useful to the literature. If you also find similar value in this research and future works that can build on these findings, we would really appreciate it if you could consider increasing your score. Thank you very much once again, and we are happy to follow up with any additional questions you may have!

---

> > ### Author Response · Authors · 2023-11-19
> > **We would love to hear from you!**
> >
> > Dear Reviewer m32R,
> >
> > Once again thank you very much for your detailed review and the time you spent. As we are nearing the end of the discussion period, we would like to ask if the questions you raised have been addressed. We hope you find our responses useful and would love to engage with you further if there are any remaining points.
> >
> > We understand that the discussion period is short, and we sincerely appreciate your time and help!

---

> > > ### Comment · Reviewer_m32R · 2023-11-30
> > > **acknowledgement**
> > >
> > > Thank for you for answering my questions!

---

### Official Review · Reviewer_KSCt · 2023-11-07

**Soundness:** 2 fair
**Presentation:** 3 good
**Contribution:** 2 fair
**Rating:** 6
**Confidence:** 4

**Summary:**

This work proposes to use Transformers’ internal attention mechanisms to understand the internal mechanism behind factual errors, in contrast to previous work focusing more on discovering internal mechanism behind generate *correct* factual answers. It poses the problem of answering factual queries as a constraint satisfaction problem whereby each factual queries presents a conjunction of constraints (e.g. “is a basketball player” and “was born in 1988”) and the query is answered correctly if each constraint is satisfied. The paper begins with a series of analyses linking attention on the constraint tokens with factuality, then proposes the SAT probe, a linear classifier over the attention on the constraint tokens, trained to predict when all constraints will be satisfied (and thus when the LLM will make factually correct predictions).

**Strengths:**

1. Casting factual queries as a constraint satisfaction problem is an interesting approach, and the specifically the finding that attention on the constraint tokens is correlated/predictive of factual correctness is novel.

2. Compared to confidence-based classification, the SAT probe is more efficient — allowing the model to not have to go through the entire inference procedure — and fine-grained — allowing us to isolate of which exact constraint was disregarded.

3. The experiments covered many different datasets and a wide range of model sizes — and the findings generally carried over between the various model sizes and datasets. The wealth of empirical findings is insightful and convincing.

**Weaknesses:**

1. My main point of contention is that the SAT probe actually underperforms confidence classification by a statistically significant margin in 17/27 of the settings studied in in Figure 6 (it would be good to note what type of error bars are being plotted and how many trials this was over/what the source of randomness was). This seems to undermine “we find that SAT PROBE mostly performs comparably to and sometimes better than the model’s CONFIDENCE in the correctness prediction task” (section 5.1, results).
    1. Section 5.2 and appendix dive into results that demonstrate that analyzing attention at earlier layers can be useful — as it is generally more efficient and allows us to interpretably isolate the constraint that failed. This should be moved up and highlighted as one of the main advantages of this method. (The preliminary analyses on popularity and constrainedness predicting performance / attention predicting popularity and constrainedness can be compressed.)
    2. Table 7 examines the ability of SAT probe to isolate *which* constraint was violated when there are multiple constraints. Some more detail is necessary here: in the dataset, when 1 constraint is violated, how often is it the case that both constraints are violated? Can we isolate the experiment to cases where one and only one of the constraints are violated, and have the SAT probe predict which one?

2. I do wonder about the generality of thinking of factuality as a CSP — in particular simply treating it as a conjunction of constraints. There are many other types of composition, such as disjunction, nesting (multi-hop queries like “parents of the President of the USA”), etc., that cannot be dealt with in this framework. Furthermore, not are all factual queries can be dealt with just simply using constraints. For example, there may be additional reasoning operations on top of constraints (e.g. numerical counting). While I don’t expect this one paper to comprehensively deal with all types of factual queries, perhaps some discussion of the coverage of this framework is warranted.
    1. More generally, prior work has already examined factual queries from the lens of database (SQL) queries, which implicitly already carries constraints (i.e. the WHERE clauses) while also being more flexible to different types of constraint compositions, and many more operations. What is the comparative advantage to thinking of factual queries as conjunctions of constraints in this way?

3. Furthermore, the name “constraint satisfaction problems” generally suggests a different set of problems to me — where the key focus is not on evaluating the factuality of each atomic variables, but on how we may search through assignments to variables in order to create a permissible solution. (This is a relatively minor naming point, though does open up a potential question on the implication of thinking of queries as a constraint satisfaction problem, in the more generic sense, which could introduce much more novelty to this paper.)

**Questions:**

1. Are there any takeaways from this analysis and method for how we can actually fix factual errors? Or perhaps prevent models from generating factual errors?

2. This method mainly relies on LM uncertainty to pick up when the LM is wrong, which is fine (I’m not sure if there’s any other way to internally discover when the LM is lying without external tools). However, broadly speaking, are there ever cases of the LM being confidently wrong? Or is it the case that, because of the distribution of the pre-training data, whenever there’s factual errors it will always be much less certain than when there’s correct answers?

3. Section 5.2, Early Stopping: “For Llama-2 7B and 13B, we can stop the inference early without degradation in the average performance and save 50% of wall-clock time on failures for most datasets.” Is there a breakdown of wall-clock time for each of the datasets?

---

> ### Author Response · Authors · 2023-11-10
>
> Hello Reviewer KSCt,
>
> First, thank you for your time reviewing our work! We believe the above review belongs to another paper, and the review for our paper is possibly elsewhere. Would you mind updating your review?
>
> Thank you so much!

---

> > ### Comment · Reviewer_KSCt · 2023-11-14
> > **Fixed review mixup**
> >
> > Hi authors, thank you for pointing this out! That was my bad. I have updated my review with the one for your paper. Apologies for the mixup!

---

> > > ### Author Response · Authors · 2023-11-15
> > > **Thank you!**
> > >
> > > Thank you for quickly updating the review and allowing us to respond in time!

---

> ### Author Response · Authors · 2023-11-15
> **Response to Reviewer KSCt 1/2**
>
> Dear Reviewer KSCt,
>
> Thank you so much for your detailed insights and the time you took to review our paper, and also for updating your review promptly, we really appreciate your help! We are extremely excited that you find our framework novel, and experiments insightful and convincing!
>
> At a high level, we provide new results where
> - (i) we show that the errors made by attention and confidence are different
> - (ii) this lets us come up with a method that combines confidence and attention, which works better than confidence or attention alone.
>
> We further incorporated the suggestions, clarified the generality of the framework compared to the analyses in the existing literature, and resolved ambiguities in the text.
>
> Below we respond to the individual points you raised.
>
> 1. *My main point of contention is that the SAT probe actually underperforms confidence classification by a statistically significant margin*, *However, broadly speaking, are there ever cases of the LM being confidently wrong?*
>
> **Confidence vs Attention:** We give two sets of new results to address this question:
>
> **Do confidence and attention result in the same predictions? Combining attention and confidence**: In Appendix E Figure 15, we compare the predictions by Attention vs Confidence for all three models in the Llama family. Overall, while they do correlate, there are many cases where both predictors disagree. For instance, the model can be overconfident and attention can correctly identify factual errors (see top left subfigure). Because the errors are not identical, we can combine them to build a better system.
>
> We extended our results with a *Combined* predictor, where we add confidence in addition to the attention features, and perform logistic regression. This model mostly performs the best overall, in terms of AUROC (Figure 6), and risk in either tail (bottom 20% and top 20% most confident predictions, Figures 9 and 10). Thus, there is value in considering both predictors, in addition to providing fine-grained predictions per individual constraints and early stopping with attention.
>
>  **Why is the finding that model attention can be used to predict constraint satisfaction and is comparable to confidence important\interesting?** We would like to highlight that the LLM is optimized using confidence as the objective (e.g. probability of the next token). We find it to be an extremely interesting finding that the process that the LLM uses to produce an output, without information about the objects it is acting on (predicted probability distribution), gives almost as much information as the output. We believe our findings could lead to several follow-up works in studying factual errors and the model’s reliability. Why do we have more attention to popular entities? How can we manipulate attention to increase constraint satisfaction and steer the model behavior? Our framework and findings open interesting questions and can pave the way for future research in this area.
>
> - *This should be moved up and highlighted as one of the main advantages of this method.*, *This seems to undermine “we find that SAT PROBE mostly performs comparably to and sometimes better*
>
> We adjusted the tone of the claims about that table per se and highlighted the advantage of attention in being more efficient and allowing us to interpretably isolate the constraint that failed. Thank you for these suggestions.
>
> - *it would be good to note what type of error bars are being plotted and how many trials this was over/what the source of randomness was*
>
> Error bars are discussed in the caption of Figure 6, i.e. “Error bars show the
> standard error over 10 different random train/test splits.”

---

> ### Author Response · Authors · 2023-11-15
> **Response to Reviewer KSCt 2/2**
>
> 2. *I do wonder about the generality of thinking of factuality as a CSP*
>
> **Generality of the Framework:** We agree with the reviewer that the CSP framework does not cover all potential queries to LLMs. Our work lays out a framework for mechanistic interpretability to analyze a broader class of queries, thus contributing to the literature: **existing work** in factuality and mechanistic interpretability in LLMs (e.g. Meng et al. NeurIPS 2022, Geva et al. EMNLP 2023) only considered single-constraint queries with a well-defined, yet restrictive (subject, relation, object) structure. We would appreciate it if the reviewer could consider that in the context of mechanistic understanding of LLMs, moving to multiple constraints is novel and allows broader investigations. While there are still limitations, it creates further opportunities regarding what we can understand with a mechanistic lens, e.g. in this case by directly mapping attention patterns of individual constraints to model performance.
>
> Thanks for pointing out the analogy to how SQL and other declarative language model constraint-based search! Indeed, previous work (such as “Conjunctive-query containment and constraint satisfaction” from Kolaitis & Vardi, 1998 but also others) have used the same framework to model similar queries, albeit not for machine learning models. The analogy is a great argument for leveraging the same framework but with an ML perspective and by using architectural information native to trained models (rather than stored information in database systems). If we missed any other works here, please let us know and we will be happy to discuss further and cite further work on this topic.
>
>
> 3. *Is there a breakdown of wall-clock time for each of the datasets?*
>
> **Wall-clock time:** Our comments are based on the fact that the inference time scales linearly in the number of layers, and we can sometimes use less than 50% of the layers to predict failure as successfully as using the entire network. While the exact inference time will depend on the prompt length and the hardware, it will scale linearly in the number of layers.
>
> 4. *Table 7 examines the ability of SAT probe to isolate which constraint was violated when there are multiple constraints. Some more detail is necessary here: in the dataset, when 1 constraint is violated, how often is it the case that both constraints are violated?*
>
> The numbers vary across datasets – sometimes only one fails and sometimes both do. Below is a table summarizing the statistics for Llama-2 70B performance in the first train/test split as an example:
>
> | Dataset Name | Fraction of Samples | Which Constraint Failed |
> |--------------|---------------------|-------------------------|
> | Nobel Winner | 0.59                | Only One                |
> | Nobel Winner | 0.26                | Both                    |
> | Movies       | 0.04                | Only One                |
> | Movies       | 0.53                | Both                    |
> | Books        | 0.17                | Only One                |
> | Books        | 0.67                | Both                    |
> | Words        | 0.57                | Only One                |
> | Words        | 0.13                | Both                    |
>
> Thus, indeed there are datasets where only one of the constraints fails, and attention can identify the ones that do (shown by Table 7).
>
> **Insights into how to fix errors:** This is an important question that warrants future research. While we do not have experiments regarding this point, our findings can lead to future investigations. E.g. It would be interesting to intervene in attention to constraints to modify the model behavior. However, our setting is akin to Selective Classification, where we focus on abstention as the action to take when reliability is expected to be low. We added further suggestions in the conclusion section on this point, thank you for the suggestions.
>
> **Overall**, we do believe we are studying the problem of factual error prediction from a novel point of view, our findings around attention and factual error prediction, datasets, and the framework could prove useful to the literature. If you also find similar value in this research and future works that can build on these findings, we would really appreciate it if you could consider increasing your score. Thank you very much once again, and we are happy to follow up with any additional questions you may have!

---

> > ### Author Response · Authors · 2023-11-19
> > **We would love to hear from you!**
> >
> > Dear Reviewer KSCt,
> >
> > Once again thank you very much for your detailed review and the time you spent. As we are nearing the end of the discussion period, we would like to ask if the questions you raised have been addressed. We hope you find our responses useful and would love to engage with you further if there are any remaining points.
> >
> > We understand that the discussion period is short, and we sincerely appreciate your time and help!

---

### Author Response · Authors · 2023-11-15
**General Updates**

We thank all reviewers for their time and insightful comments. We are very excited about the reviewers' various positive comments: "The wealth of empirical findings is insightful and convincing"[KSCt], "It may inspire future researchers to discover and probe more interpretable inner patterns of LLMs, which could potentially help researchers understand and explain models’ behaviours."[MD6P], "This work looks at mechanistic interpretability especially with respect to factor errors made by LLMs, which has not been studied before as per my knowledge"[m32R], "..attention on the constraint tokens is correlated/predictive of factual correctness is novel."[KSCt].

We updated our paper considering the points raised in the reviews, and below we give general updates and responses to common questions. We also highlighted the changes in red in the paper.

**1. Confidence vs Attention:** The reviewers raised multiple questions about how Attention compares to Confidence. Our existing experiments demonstrate two main unique benefits of attention over using confidence, which are i) providing fine-grained feedback for individual constraints (confidence only provides a single value for the entire query), and ii) early stopping to save computation (confidence uses one full forward pass). We provide two sets of new experimental results to further demonstrate the utility of attention, and how it brings additional value:

**Do confidence and attention result in the same predictions? Combining attention and confidence**: We compare the predictions and errors made by Attention vs Confidence for all three models in the Llama family in Appendix E Figure 15. Overall, while they do correlate, there are many cases where both predictors disagree. For instance, the model can be overconfident and attention can correctly identify factual errors (see the top left subfigure). Because the errors are not identical, this indicates we can combine them to build a better system.

We extended our results with a *Combined* predictor, where we add confidence in addition to the attention features, and perform the same logistic regression. This model mostly performs the best overall, in terms of AUROC (Figure 6), and risk in either tail (bottom 20% and top 20% most confident predictions, Figures 9 and 10). Thus, there is value in considering both predictors and attention does add value.

**Why is the finding that model attention can be used to predict constraint satisfaction and is comparable to confidence important or interesting?** We would like to highlight that the LLM is optimized using confidence as the objective (e.g. probability of the next token). We find it to be an extremely interesting finding that the process in which the model generates an output alone (e.g. attention values) is comparable to the output itself, confidence, and believe our findings could lead to several follow-up works in studying factual errors and the model’s reliability. Why do we have more attention to popular entities? How can we manipulate attention to increase constraint satisfaction and steer the model behavior? Our framework and findings open interesting questions and can pave the way for future research in this area.

**2. Generality of the framework:** We agree with the reviewers that the CSP framework does not cover all potential queries to LLMs. We highlight that the *existing published work* in factuality in LLMs (e.g. Meng et al. NeurIPS 2023, Geva et al. EMNLP 2023) only considered single-constraint queries with well-defined, yet restrictive (subject, relation, object) structure, and mostly built on only the CounterFact dataset. Our work lays out a framework for mechanistic interpretability to model a broader class of queries, thus contributing to the literature. Further, we do not only look at CounterFact and we expand the set of datasets further to questions that are verifiable and that can also be reused in future research. While there are still limitations, it creates further opportunities regarding what we can understand with a mechanistic lens, in particular by directly mapping attention patterns of individual constraints to model performance.  We added further discussion on this in the conclusion section, thanks to all reviewers for their suggestions.

---

### Meta-Review · Area_Chair_xn1F · 2023-12-07

**Metareview:**

This paper puts forward the novel idea of viewing factual queries as a constraint satisfaction problem which offers  an interesting angle on  on handling factual correctness in LLMs. This approach, particularly the finding that attention on constraint tokens correlates with factual correctness deserves recognization and the paper meets the bar for publication.  The experiments are thorough and the writing, figures, and overall presentation are high quality. One of the concerns I have pertains to what appears to be limited applicability.  Additionally, I am concerned about the use of the term "SAT" as it may lead to confusion due to its established meaning in other contexts. I suggest that the authors consider renaming this to avoid potential misunderstanding. While this is a minor point, it could significantly impact clarity and understanding. I would suggest the authors find another name - this is minor but might not be small in terms of its effect on the work's impact.

**Justification For Why Not Higher Score:**

One of the concerns I have pertains to what appears to be limited applicability: the approach's focus on factual queries as constraint satisfaction problems raises questions about its applicability to a broader range of factual errors, like those involving numerical counting or multi-hop reasoning. The method's reliance on the availability of constraint entities also limits its practicality

**Justification For Why Not Lower Score:**

The paper has a novel conceptual idea, thorough experiments, and presentation is of high quality.

---

### Decision · Program_Chairs · 2024-01-16

Accept (poster)